# Targeting mutant p53-expressing tumours with a T cell receptor-like antibody specific for a wild-type antigen

Lionel Low[1], Angeline Goh[1], Joanna Koh[2], Samantha Lim[2] & Cheng-I Wang[1]*

Accumulation of mutant p53 proteins is frequently found in a wide range of cancers. While conventional antibodies fail to target intracellular proteins, proteosomal degradation results in the presentation of p53-derived peptides on the tumour cell surface by class I molecules of the major histocompatibility complex (MHC). Elevated levels of such p53-derived peptide-MHCs on tumour cells potentially differentiate them from healthy tissues. Here, we report the engineering of an affinity-matured human antibody, P1C1TM, specific for the unmutated p53$_{125-134}$ peptide in complex with the HLA-A24 class I MHC molecule. We show that P1C1TM distinguishes between mutant and wild-type p53 expressing HLA-A24$^+$ cells, and mediates antibody dependent cellular cytotoxicity of mutant p53 expressing cells in vitro. Furthermore, we show that cytotoxic PNU-159682-P1C1TM drug conjugates specifically inhibit growth of mutant p53 expressing cells in vitro and in vivo. Hence, p53-associated peptide-MHCs are attractive targets for the immunotherapy against mutant p53 expressing tumours.

---

[1] Singapore Immunology Network, Agency for Science, Technology and Research, 8A Biomedical Grove, Singapore 138648, Singapore. [2] School of Life Sciences and Chemical Technology, Ngee Ann Polytechnic, 535 Clementi Road, Singapore 599489, Singapore. *email: Wang_Chengl@immunol.a-star.edu.sg

The p53 transcription factor plays an important role in response to cellular stress and is crucial in the protection against cancer development. Through the regulation of genes involved in DNA repair, cell cycle arrest and apoptosis, p53 ensures genetic integrity by preventing the accumulation of aberrations that would otherwise lead to malignancy and oncogenesis. In the absence of stress signals, p53 is kept at low levels by the continuous ubiquitination by E3 ubiquitin ligases such as MDM2, and degradation by the proteasome. Stress signals result in post-translational modifications of the p53 protein, leading to the increased stability of the p53 protein by disrupting the p53-MDM2 interaction, and the subsequent activation of p53[1].

The *p53* gene is the most commonly mutated gene found in human malignancies. While frameshift and nonsense mutations have been observed, missense mutations resulting in single amino acid changes in the DNA-binding domain make up the majority of tumour-associated mutations. Studies have further identified six "hotspot" positions in the DNA-binding domain at Arg175, Gly245, Arg248, Arg249, Arg273 and Arg282 that are the most frequently mutated[2]. These mutations are known to increase the stability of the mutant proteins and also disrupt the native conformation of the p53 protein, resulting in the inability to recognize and bind the cognate p53 response elements, while suppressing wild-type p53 and other p53 family members[3–5], and thus impairing tumour-suppressive function and promoting oncogenesis.

CD8$^+$ T cells recognize short peptide epitopes presented on the cell surface of tumour cells in complex with a class I protein of the major histocompatibility complex (MHC) via their T cell receptors (TCRs). Proteins expressed by the tumour cells are continuously degraded and presented as a peptide-MHC (pMHC) antigen to stimulate anti-tumour CD8$^+$ T cell responses[6]. The ability to target such pMHCs has been achieved by soluble TCRs or antibodies with TCR-like recognition, termed TCRL (TCRL) or TCR mimic antibodies, with great therapeutic potential[7–15]. Elevated p53 levels in tumours expressing mutant p53 may result in higher levels of presentation of p53-derived peptides by MHC molecules. Peptides containing mutant sequences are rare due to the MHC-binding restrictions; however, elevated levels of MHCs presenting wild-type p53 peptide sequences can potentially differentiate malignant expressing mutant p53 from healthy cells expressing wild-type p53[16–18].

Here, we report the engineering of a TCRL antibody, P1C1TM, specific for a wild-type p53$_{125–134}$ peptide presented by the HLA-A24:02 (HLA-A24) MHC allele[17]. We show that P1C1TM can differentiate between mutant and wild-type p53-expressing HLA-A24$^+$ cell lines based on the differences in the antigen expression level. Its implications and potential applications for cancer therapy are discussed.

## Results

### Isolation of p53$_{125–134}$/HLA-A24-specific antibodies.
A human Fab library consisting of $3 \times 10^{10}$ M13 phagemids[19] were used for the isolation of p53$_{125–134}$/HLA-A24-specific antibodies. Negative selection against a control pMHC and streptavidin beads was done prior to positive selection to reduce non-specific clones. After three rounds of biopanning, 36 single Fab clones were selected based on their specific binding to p53$_{125–134}$/HLA-A24 over the control pMHC in an enzyme-linked immunosorbent assay (ELISA). DNA fingerprinting and subsequent sequencing identified four unique clones, P1H4, P1B11, P1A8 and P1C1. The four clones were expressed in immunoglobulin G1 (IgG1) form and assessed for their specificities to the p53$_{125–134}$/HLA-A24 pMHC by ELISA. Clones P1H4 and P1C1 showed the strongest

binding to p53$_{125–134}$/HLA-A24 pMHC, but P1C1 showed the least non-specific binding to the control pMHC (Fig. 1a).

### Characterization of TCRL antibody P1C1.
The binding specificity of clone P1C1 was further characterized using an HLA-A24$^+$ but p53-null SaoS2 cell line. Cells were either unpulsed or pulsed with various known HLA-A24-restricted peptides and stained with $10 \, \mu g \, mL^{-1}$ of antibody. P1C1 showed insignificant binding to unpulsed SaoS2 cells, while strong binding was observed only to p53$_{125–134}$-pulsed SaoS2 but not to control peptides (Fig. 1b). SaoS2 cells pulsed with varying concentrations of p53$_{125–134}$ peptide showed that P1C1 bound cells pulsed with at least 400 nM of peptides (Fig. 1c). Next, SaoS2 cells pulsed with $10 \, \mu M$ peptides were stained with varying concentrations of P1C1 (Fig. 1d). P1C1 staining was observed at an antibody concentration of at least $10 \, ng \, mL^{-1}$.

To reduce potential immunogenicity of the lead P1C1 antibody, site-directed mutagenesis was performed to convert the backbone to germline-like sequences (IGHV4−31*03, IGLV1-40*01) (Supplementary Fig. 1a). The germline P1C1, P1C1gl, demonstrated similar binding profile as the parental P1C1 antibody (Supplementary Fig. 1b). Lastly, the ability of P1C1 to recognize and bind HLA-A24$^+$ HT29 cells that express high levels of mutant p53 was examined. However, minimal staining was seen even with $10 \, \mu g \, mL^{-1}$ of antibodies (Fig. 1e). Together, the data suggested that P1C1 bound with relatively low affinity.

### Affinity maturation of P1C1.
A potential challenge to the use of soluble TCRs or TCRL antibodies is the relatively low number of specific pMHC complexes available for binding per cell as compared to other traditional antibody targets, for example, HER2[20]. Hence we proceeded to improve the affinity of P1C1 by generating four separate libraries in which the complementarity-determining regions (CDRs) 1 and 3 of the heavy and light chains were randomized with degenerate codons encoding for the wild-type amino acid and a restricted diversity that includes primarily Ala, Ser, Tyr and Asp (Table 1)[21]. Glycine and tryptophan residues were left unchanged to preserve structural integrity[22]. The constructed CDR1$_{heavy}$, CDR3$_{heavy}$, CDR1$_{light}$, and CDR3$_{light}$ libraries contained $6.7 \times 10^7$, $4.5 \times 10^7$, $3.8 \times 10^8$ and $1.4 \times 10^8$ clones, respectively, covering the theoretical maximum diversity of each library of 256, $1.6 \times 10^5$, $3.8 \times 10^8$ and $1.6 \times 10^7$, respectively. Individual clones from the CDR1$_{heavy}$ library were directly screened for specificity by ELISA due to its small theoretical diversity, while the CDR3$_{heavy}$, CDR1$_{light}$ and CDR3$_{light}$ were subjected to three rounds of biopanning with increased stringency to enrich for stronger binders. While no improved binders were isolated from the CDR3$_{light}$ library, 28, 15 and 11 unique and dominant clones were identified from CDR1$_{heavy}$, CDR3$_{heavy}$ and the CDR1$_{light}$ libraries, respectively, and were assessed by bio-layer interferometry for improved off-rate. Subsequently, clones 2E3, 1E11 and 1G7 were identified as the best binders from the CDR1$_{heavy}$, CDR3$_{heavy}$ and the CDR1$_{light}$ libraries, respectively.

### Characterization of P1C1 affinity-matured mutants.
Clones 2E3, 1E11 and 1G7 and a triple mutant (P1C1TM) incorporating the mutations in the three CDRs (Fig. 2a) were expressed in IgG1 format and their binding affinities were analysed by ELISA. 1E11 showed the biggest improvement in binding, while 2E3 and 1G7 exhibited similar binding affinity as P1C1gl (Fig. 2b). Interestingly, P1C1TM had significantly higher binding affinity, measured at around 5 nM (Fig. 2c), suggesting that the mutations in the individual CDRs had a synergistic effect when combined. Affinity measurements of the P1C1gl antibody ($K_D = 116 \, nM$) and the single mutants confirmed that 1E11 (20 nM) had the

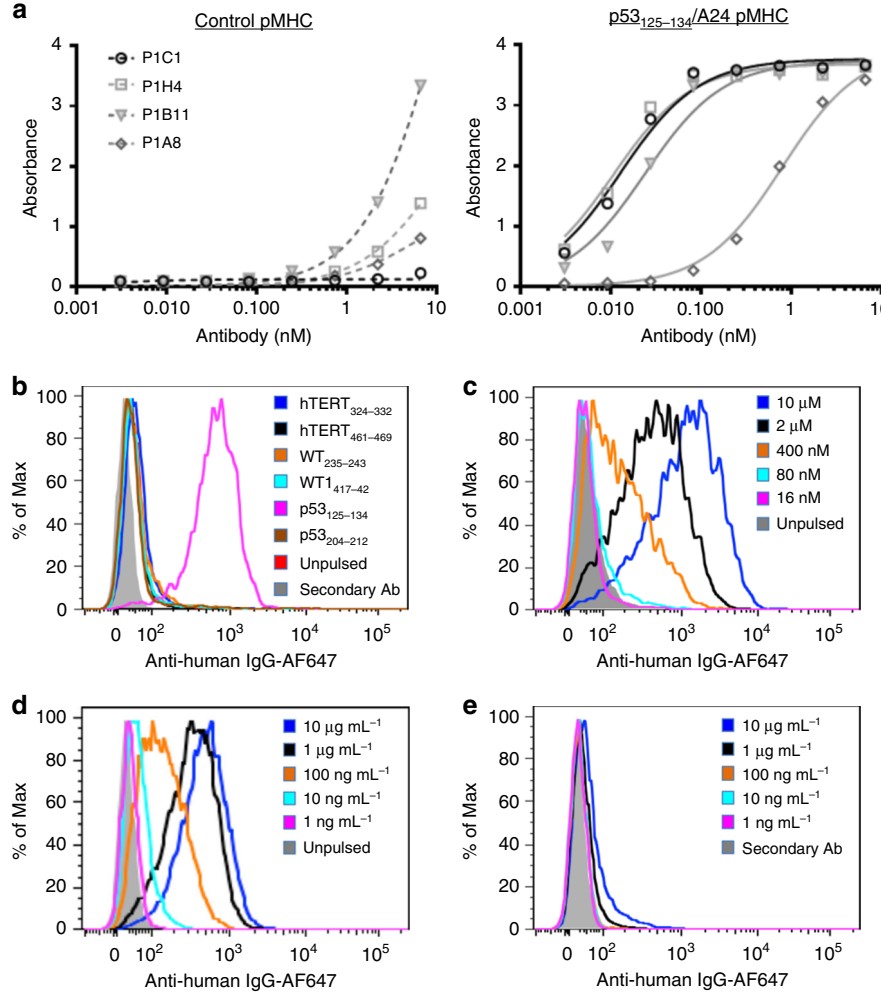

**Fig. 1** Identification of TCRL antibody P1C1 specific for the p53$_{125-134}$/A24 pMHC. **a** Binding specificity and avidity of four leads, P1C1, P1H4, P1B11 and P1A8, to a control hTERT$_{461-469}$/A24 pMHC (left) and the target p53$_{125-134}$/A24 pMHC was analysed by ELISA. **b** A24$^+$, p53-null SaoS2 cells pulsed with 10 μM six known A24-restricted peptides were stained with 10 μg mL$^{-1}$ of P1C1 antibodies. Staining was observed only with cells pulsed with the p53$_{125-134}$ peptide. P1C1 binding was further analysed by **c** staining SaoS2 cells pulsed with a range of p53$_{125-134}$ peptide concentrations or **d** staining 10 μM p53$_{125-134}$ peptide-pulsed SaoS2 cells with a range of antibody concentrations. **e** A24$^+$, mutant p53$^{R273H}$-expressing HT29 cells were stained with a range of antibody concentrations of P1C1. Data are representative of two or more independent experiments.

most significant improvement in binding affinity as compared to 2E3 (69 nM) and 1G7 (66 nM) (Supplementary Fig. 2).

The specificity of P1C1TM was next analysed by flow cytometry. SaoS2 cells were pulsed with a panel of HLA-A24-restricted peptides and stained with 10 μg mL$^{-1}$ of antibody. The specificity of the affinity-matured clone was preserved as staining was only observed when cells were pulsed with p53$_{125-134}$ (Fig. 2d). SaoS2 cells were then pulsed with a range of concentration of p53$_{125-134}$ peptides and stained with 10 μg mL$^{-1}$ of antibody. As compared to the parent P1C1 clone, staining by P1C1TM was observed when cells were pulsed with peptides at as low as 16 nM (Fig. 2e).

**Fine specificity of P1C1TM**. To better understand the specificity and possible cross-reactivity of P1C1 and importantly the affinity-matured P1C1TM, the fine specificity of P1C1gl and P1C1TM was first assessed by alanine scanning mutagenesis of the p53$_{125-134}$ peptides. Position 5 was left unchanged, as it is an alanine in the wild-type sequence. Figure 3a shows that alanine substitutions at positions 3, 4, 6 and 8 significantly reduced the binding of P1C1 compared to wild-type p53$_{125-134}$ peptide-pulsed cells. Reduction of P1C1 binding observed was not due to reduced

peptide binding to the HLA-A24 molecule itself as the mutant peptides, except P2A and P10A, were still able to rescue HLA-A24 complexes in a ultraviolet (UV)-peptide exchange HLA-stability ELISA (Supplementary Fig. 3a). Positions 2 and 10 are known anchor residues that are important for the peptide's binding to HLA-A24[23]; thus, substitutions to alanine reduced their ability to form stable pMHC and thus also abolished P1C1gl binding. Affinity-matured P1C1TM exhibited reduced sensitivity to the positions 3, 4 and 6 but not to position 8 (Fig. 3a). This suggests that the CDR loops targeted in our affinity maturation strategy play a role in binding the central residues of the p53$_{125-134}$ peptide.

We next looked at possible cross-reactive epitopes in other proteins encoded in the human genome. A motif search of the Kyoto Encyclopaedia of Gene and Genomes (KEGG) database followed by HLA-A24-binding affinity prediction by NetMHC 3.0 identified several potential cross-reactive peptides (Fig. 3b). The ability of these peptides, as well as the murine homologue p53$_{119-128}$ that differs only at positions 5 and 9, to be presented by HLA-A24 was assessed by UV-peptide exchange HLA-stability ELISA. All 15 peptides tested rescued cleaved UV-cleavable HLA-A24 pMHCs to similar or better extent than the p53$_{125-134}$

**Table 1 Degenerate codons and their corresponding amino acid sequences of CDR libraries designed for affinity maturation.**

| CDR | | | | | | | |
|---|---|---|---|---|---|---|---|
| **Heavy CDR1** | | | | | | | |
| Original seq. | S | Y | W | Y | G | F | S |
| Degenerate codon | KMT | KMT | TGG | KMT | GGT | KHT | KMT |
| Encoded amino acid | Y/A/D/S | Y/A/D/S | W | Y/A/D/S | G | Y/A/D/S/F/V | Y/A/D/S |
| **Heavy CDR3** | | | | | | | |
| Original seq. | E | A | G | N | D | H | |
| Degenerate codon | KMW | KMT | GGA | DMT | KHT | BMT | |
| Encoded amino acid | Y/A/D/S/E | Y/A/D/S | G | Y/A/D/S/N/T | Y/A/D/S/F/V | Y/A/D/S/H/P | |
| **Light CDR1** | | | | | | | |
| Original seq. | T | S | S | S | I | D | |
| Degenerate codon | DMT | KMT | GGG | KMT | DHT | KMT | |
| Encoded amino acid | Y/A/D/S/N/T | Y/A/D/S | G | Y/A/D/S | Y/A/D/S/D/T/I/V/F | Y/A/D/S | |
| **Light CDR3** | | | | | | | |
| Original seq. | Q | D | N | L | S | V | |
| Degenerate codon | BMW | DMT | DMT | BHT | KMT | KHT | |
| Encoded amino acid | Y/A/D/S/Q/H/P/E | Y/A/D/S | Y/A/D/S/N/T | Y/A/D/S/L/H/P/V/F | Y/A/D/S | Y/A/D/S/F/V | |

Diversity of CDR1 and CDR3 libraries of the heavy and light chains of P1C1. Residues of the respective CDR loops were identified by Kabat definition and randomized with degenerate codons encoding for the original amino acid or a limited diversity consisting of Ala, Ser, Tyr or Asp. Gly and Typ were left unchanged to preserve structural integrity. Theoretical diversity of the heavy-chain CDR1, 3 and light-chain CDR1, 3 libraries are calculated to be 256, $1.6 \times 10^5$, $3.8 \times 10^8$ and $1.6 \times 10^7$

peptide (Supplementary Fig. 3b); thus, all peptides are able to form stable HLA-A24 pMHCs. However, P1C1TM binding was only detected with the cognate $p53_{125-134}$/A24 pMHC but not with any of the predicted cross-binders (Fig. 3c). Similarly, P1C1TM binding was seen only with SaoS2 cells pulsed with 10 µM of $p53_{125-134}$ peptides, but not with the other peptides (Fig. 3d). Hence, P1C1TM exhibits a high specificity for the $p53_{125-134}$ peptide/HLA-A24 pMHC.

**Detection of endogenously processed $p53_{125-134}$/A24 pMHC.** Next, we studied and compared the physiological expression and levels of $p53_{125-134}$/A24 pMHC complexes on tumour cell lines with various p53 statuses, using the affinity-matured P1C1TM. Three HLA-A24[+] cell lines, the osteosarcoma SaoS2 (p53[null]), hepatocellular carcinoma HepG2 (p53[wt]) and colon adenocarcinoma HT29 (p53[R273H]) express different levels of p53 (Supplementary Fig. 4). Staining of these three cell lines with P1C1TM show that levels of $p53_{125-134}$/A24 pMHCs detected by P1C1TM correlated with the level of p53 expression (Fig. 4a). To further validate this, several other cell lines with various p53 statuses and expression levels (Supplementary Fig. 4) were transduced with HLA-A24. The breast cancer cell lines MDA-MB-231 and BT474 express mutant p53, while the lung cancer cell line A549 and the breast cancer cell line MCF7 express wild-type p53. Staining of these cell lines with P1C1TM showed that $p53_{125-134}$/A24 pMHCs levels were higher in cells expressing mutant p53 than cells expressing wild-type p53 (Fig. 4b). Thus, our results suggest that the differential levels of $p53_{125-134}$/A24 pMHC complexes potentially allows P1C1TM to distinguish between tumour cells expressing mutant p53 and healthy cells expressing wild-type p53.

To further assess binding of P1C1TM of healthy cells expressing wild-type p53, we purified peripheral blood mononuclear cells (PBMCs) from healthy donors and stained the total PBMCs with P1C1TM. Similar to the T1-116C antibody reported by Li et al.[8], staining of P1C1TM was not observed in both A24[+] and A24[−] PBMCs from various donors (Fig. 4d and Supplementary Fig 5a.). Staining of A24[+] PBMCs was observed only when PBMCs were pulsed with the $p53_{125-134}$ peptide (Fig. 4d). However, upregulation of p53 has been reported in activated T lymphocytes[24,25]. Indeed, in our hands, we found that T cells activated with anti-CD3 and anti-CD28 agonists showed an increase in intracellular p53 (Supplementary Fig. 4c). Subsequently, staining by P1C1TM was observed only in the activated A24[+] T cells but not in resting A24[+] T cells or A24[−] T cells (Fig. 4e). This was further shown in four other A24[+] and two other A24[−] donors (Supplementary Fig. 5b).

Next, we treated HT29, A549-A24 and MCF7-A24 cells with 100 U mL$^{-1}$ interferon-γ, a cytokine known to upregulate the MHC class I expression and the components of the antigen-processing pathways[26]; 5 µM nutlin, a small molecule that inhibits the Mdm2-p53 interaction or 5 µM of the proteasome inhibitor MG132 (bortezomib). Changes in levels of HLA-A24, intracellular p53 and $p53_{12-134}$/A24 complexes were then monitored after treatment (Supplementary Fig 6). Expectedly, levels of HLA-A24 detected on the surface increased significantly in HT29 cells treated with interferon-γ, resulting in a similar increase in P1C1TM staining. Expressions of the HLA-A24 heavy-chain molecule are constitutive in both the transduced A549-24 and MCF7-A24 cells and thus are not expected to be regulated by interferon-γ directly. However, A24 expression significantly increased in the A549-A24 cells upon interferon-γ treatment accompanied by an observed increase in P1C1TM staining (Supplementary Fig. 6b). This was less evident in MCF7-A24 cells, as both A24 and P1C1TM staining were only modestly

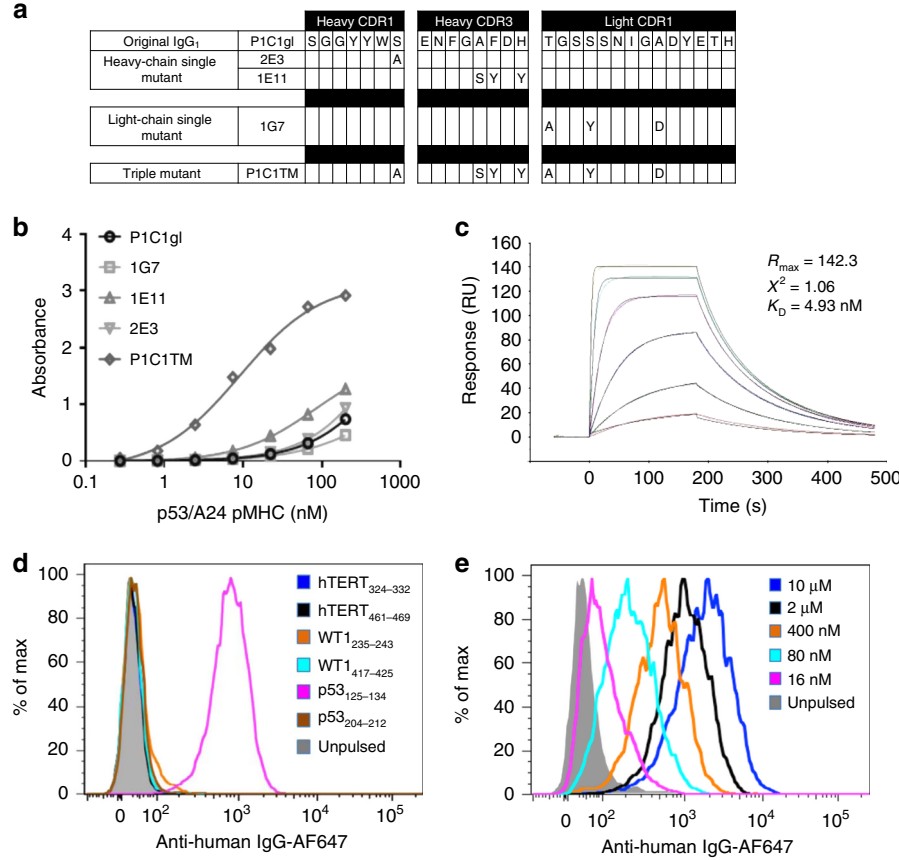

**Fig. 2** Affinity maturation of TCRL antibody P1C1. **a** CDR1 and 3 of the heavy chain and CDR1 of the light chain of the original P1C1gl clone and the affinity-matured clones are shown. Mutants 2E3, 1E11 and 1G7 were identified by off-rate comparison for improved binding from the heavy-chain CDR1, CDR3 and light-chain CDR1 libraries, respectively. Mutations were sequenced and combined in a single triple mutant, P1C1TM. **b** Comparison of the binding affinities of the germline sequence converted P1C1 (P1C1gl), individual mutants and triple mutant (P1C1TM) was done by ELISA. Soluble biotinylated recombinant p53$_{125-134}$/A24 pMHCs were incubated with immobilized antibodies and detected with HRP-conjugated streptavidin. **c** Binding kinetics of P1C1TM was analysed by surface plasmon resonance. Soluble recombinant p53$_{125-134}$/A24 pMHCs was flowed over P1C1TM antibodies captured on an anti-human IgG-coated sensor chip at a range of concentrations between 200 and 2 nM. Binding specificity and avidity of P1C1TM to p53$_{125-134}$/A24 pMHCs on cells were evaluated with SaoS2 cells pulsed with either a panel of six known A24 peptides, including p53$_{125-134}$ (**d**) or a range of p53$_{125-134}$ peptide concentrations (**e**). Data are representative of two or more experiments.

increased after treatment with interferon-γ. No increase in intracellular p53 was observed in all the treated cells.

HT29 cells treated with nutlin did not result in any change in the level of intracellular p53 or surface p53 peptide/MHC complexes (Supplementary Fig. 6b). Interestingly, MCF7-A24 cells treated with nutlin resulted in almost 2-fold increase in intracellular p53, but no significant increase was seen in surface p53 peptide/MHC complexes. Treatment with MG132 and other proteasome inhibitors is known to result in significant reductions in antigen presentation[27,28]. Here, we saw that treating HT29 cells with MG132 resulted in lower levels of HLA-A24. This corresponded to similar reductions in the levels of p53$_{125-134}$/A24 complexes detected by P1C1TM. MG132 treatment of MCF7-A24 cells resulted in significant increase in intracellular p53, but a decrease in surface A24, resulting in a net decrease in presentation of p53$_{125-134}$/A24 complexes (Supplementary Fig. 6b). No significant differences were observed in A549-A24 cells treated with either nutlin or MG132.

Lastly, to assess the specificity of P1C1TM for tumour cells expressing mutant p53 in vivo, xenografts of either MDA-MB-231 and HT29 or SaoS2 and HT29 cells were introduced into the opposite flanks of immunodeficient NOD-scid IL2rg$^{null}$ (NSG) mice (Supplementary Fig. 7a, b). After tumour establishment, mice were injected with 50 μg of either Alexa Fluor 680-conjugated

P1C1TM or human IgG$_1$ isotype control. At 48 h after injection, accumulation of the fluorescent antibodies can be observed in both the P1C1TM- and isotype control-treated mice. However, 5 days after injection, clearance of the non-specific isotype control was observed, while P1C1TM persisted in the targeted HT29 tumour, but at significantly lower levels in the control tumours (Supplementary Fig. 7a, b).

**Antibody-dependent cellular cytotoxicity potential of P1C1TM.** We evaluated P1C1's ability to mediate antibody-dependent cellular cytotoxicity (ADCC) against mutant p53 expressing tumour cells. Both P1C1gl and P1C1TM were able to mediate ADCC of unpulsed HT29 cells in a dose-dependent manner (Fig. 5a). However, while no significant cytotoxicity was observed in the control MDA-MB-231 cell line, toxicity was also low in the HLA-A24 transduced MDA-MB-231 (231-A24) cell line (Fig. 5b). The efficiency of ADCC is dependent on the level of antibodies bound to target cells[29]; thus, the lack of cytotoxicity observed in the 231-A24 cell line may be due to the relatively lower level of presentation of p53$_{125-134}$/A24 pMHC complexes (Fig. 4). This is further confirmed when enhanced P1C1gl- and P1C1TM-mediated ADCC was observed after HT29 and 231-A24 cells were pulsed with 10 μM peptides (Fig. 5a, b).

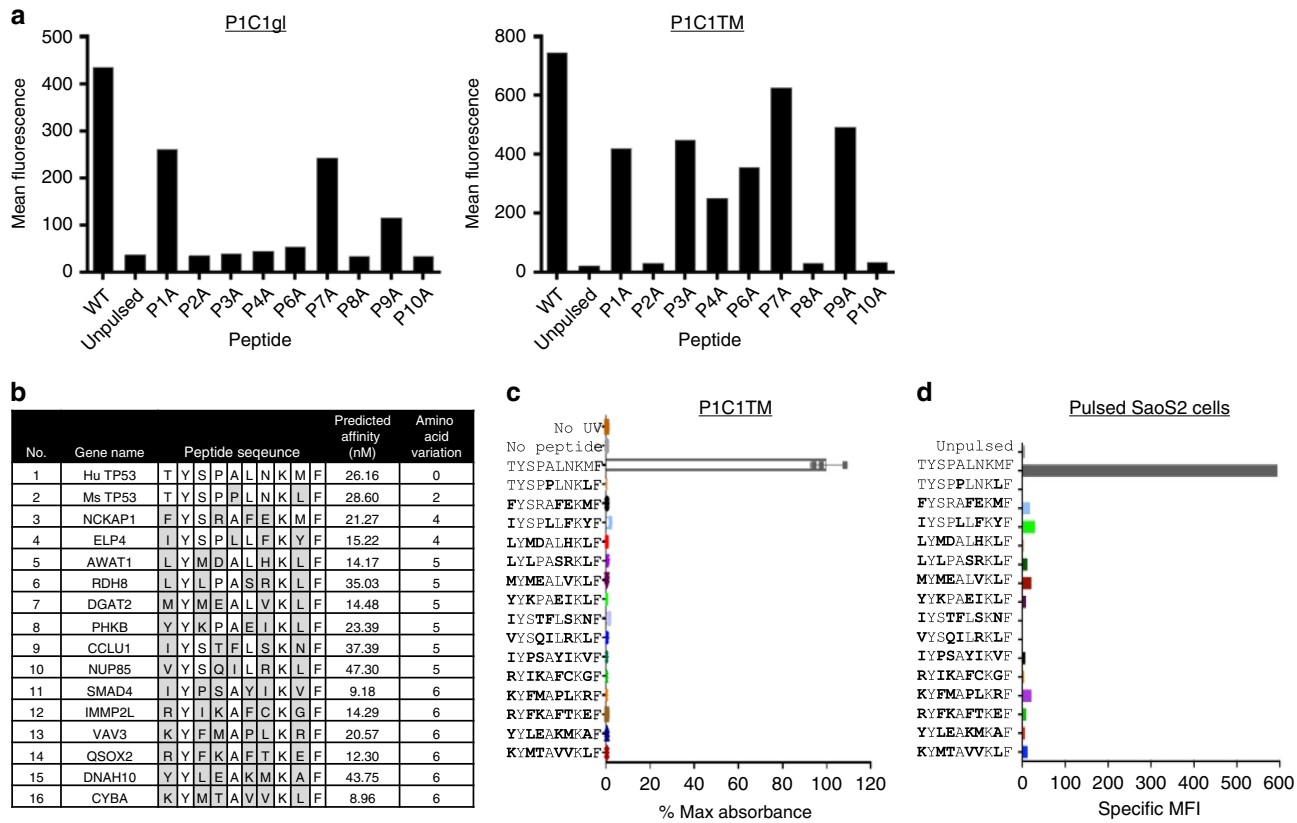

**Fig. 3** Fine specificity of affinity matured TCRL antibody P1C1TM. **a** SaoS2 cells pulsed with $p53_{125-134}$ peptide analogues containing single amino acid mutations to alanine were stained with $10 \, \mu g \, mL^{-1}$ of P1C1gl (left) or the affinity-matured P1C1TM (right) and detected with an Alexa Fluor 647-conjugated anti-human IgG secondary antibody. Data are representative of two independent experiments. **b** Potential off-target peptides were identified using the MOTIF search tool and analysed by NetMHC 3.0 for predicted binding affinity to HLA-A*24:02. Peptides with predicted binding affinity of 100 nM or less were used to determine the cross-reactivity of P1C1TM. The murine $p53_{118-127}$ that differs from the human $p53_{125-134}$ by only two amino acids at positions 5 and 9 was also included. **c** Soluble pMHCs presenting the panel peptides was produced by UV exchange and used to assess P1C1TM's fine specificity by ELISA. No UV exchange pMHC and no peptide exchange controls were used as negative controls, while pMHC UV-exchanged with $p53_{125-134}$ peptide was used as a positive control. Data are means of triplicates ± SEM. **d** The panel of peptides were pulsed on SaoS2 cells and stained with $10 \, \mu g \, mL^{-1}$ P1C1TM. Unpulsed SaoS2 cells were used as negative control, while SaoS2 cells pulsed with $p53_{125-134}$ peptide was used as a positive control. Data are representative of two independent experiments.

The substitutions of leucine 234 and leucine 235 to alanines (LALA) in the antibody Fc region have been known to reduce the binding affinity to Fcγ receptors, resulting in poorer ADCC[30]. Introduction of the LALA mutations to P1C1TM (P1C1TM-LALA) showed unchanged binding properties (Fig. 5c), but cytotoxicity was significantly reduced (Fig. 5d). Thus, these data show that the TCRL antibody P1C1TM can facilitate immune cell-mediated cytotoxicity on cells presenting high levels of $p53_{125-134}$/A24 pMHCs.

**In vitro growth inhibition by P1C1TM antibody–drug conjugates.** We next explored the potential use of P1C1TM as an antibody–drug conjugate (ADC) to deliver a cytotoxic payload to tumours expressing mutant p53. First, P1C1TM was conjugated with a pH-dependent dye pHrodo Red to evaluate the kinetics of internalization. HT29 cells were incubated with the pHrodo Red-conjugated P1C1TM either on ice or at 37 °C. The fluorescence intensity of the conjugated dye was indicative of the level of internalization of P1C1TM. Figure 6a shows that P1C1TM was internalized as rapidly as 30 min after incubation at 37C°, whereas no internalization of P1C1TM was observed when cells were incubated on ice. Next, an indirect killing assay was carried out using four different secondary ADCs. HT29 cells were first incubated with P1C1TM before equimolar of secondary

antibodies conjugated to either tubulin inhibitor monomethyl auristatin E (MMAE), DNA-alkylating agents PNU-159682 and pyrrolobenzodiazepine (PBD) or the RNA polymerase inhibitor α-amanitin (AAMT) were added. Cytotoxicity to HT29 cells was observed when cells were treated with DNA-alkylating agents, but not when cells were with treated with AAMT or MMAE. Non-specific toxicity to HT29 cells was observed when treated with highest dose of $10 \, \mu g \, mL^{-1}$ of PNU-159682, but not PBD-conjugated secondary antibody. However, >50% cytotoxicity was observed with $0.1 \, \mu g \, mL^{-1}$ PNU-conjugated secondary antibodies, while similar levels of cytotoxicity were observed only with $10 \, \mu g \, mL^{-1}$ of PBD-conjugated secondary antibodies (Fig. 6b).

P1C1TM was subsequently conjugated with PNU-159682 (P1C1TM-PNU). A binding comparison of P1C1TM and the ADC P1C1TM-PNU showed that the conjugation did not affect P1C1TM's specificity (Fig. 6c). A panel of HLA-A24-expressing cell lines incubated with either P1C1TM or P1C1TM-PNU were assessed for viability after 72 h. The cell lines expressing wild-type p53 experienced toxicity only when treated with a high concentration of P1C1TM-PNU. However, mutant p53-expressing cell lines HT29 and MDA-MB-231-A24 appeared to be much more sensitive to P1C1TM-PNU with toxicity observed with doses as low as $0.1 \, \mu g \, mL^{-1}$.

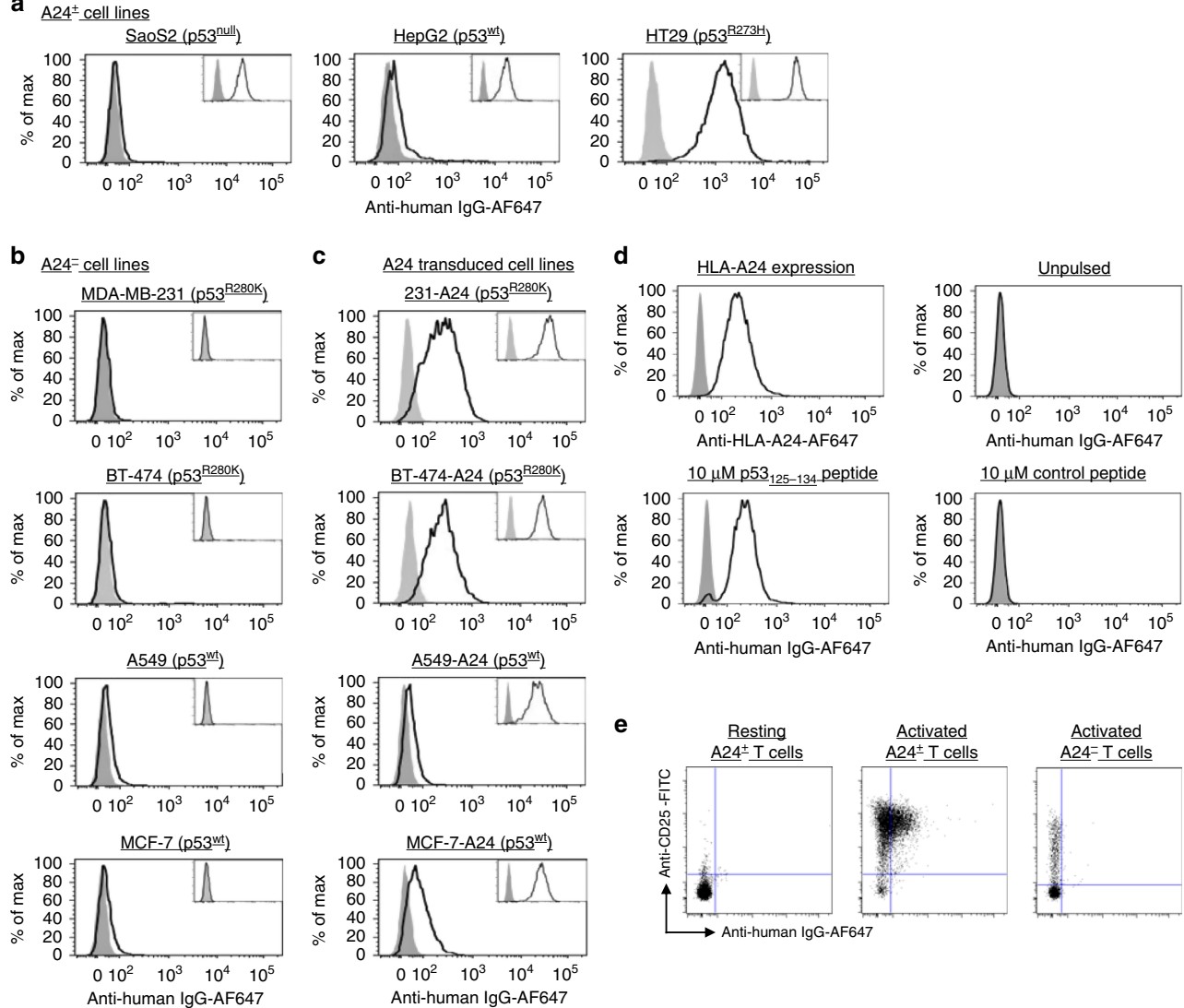

**Fig. 4** Binding of endogenously processed p53$_{125-134}$/A24 pMHCs on cells expressing wild-type and mutant p53. **a** Colon carcinoma cell line HT29 expressing HLA-A24 and mutant p53$^{R273H}$, osteosarcoma cell line SaoS2 expressing HLA-A24 with a null mutation of the *p53* gene, while hepatocellular carcinoma cell line HepG2-expressing HLA-A24 and wild-type p53 were stained with 10 μg mL$^{-1}$ P1C1TM. Significant staining was observed only with HT29 cells and minimal staining or no staining was seen with HepG2 and SaoS2 cells, respectively. HLA-A24-negative breast cancer cell lines MDA-MB-231, BT474 and MCF7, and lung carcinoma cell line A549 were transduced with HLA-A24. Staining of the untransduced (**b**) and transduced (**c**) cells with P1C1TM showed significantly higher levels of p53$_{125-134}$/A24 pMHCs on the surface of cells expressing both HLA-A24 and mutant p53 (MDA-MB-231 and BT474) compared to cells expressing wild-type p53 (A549 and MCF7). Insets show P1C1TM staining of the respective cell lines pulsed with 10 μM of p53$_{125-134}$ peptides, indicating successful transduction. Data are representative of three independent experiments. **d** P1C1TM was used to stain HLA-A24$^+$ PBMCs. No significant binding was observed in unpulsed PBMCs. P1C1TM staining was observed when pulsed with 10 μM p53$_{125-134}$ peptides but not control peptides. **e** Staining of P1C1TM was observed only in activated HLA-A24$^+$ T cells, but not resting HLA-A24$^+$ T cells or activated HLA-A24$^-$ T cells.

As activated T cells present elevated levels of p53-derived peptide-MHC complexes, we next looked at the effect of P1C1TM-PNU on activated T cells. At the highest dose tested (3 μg mL$^{-1}$), P1C1TM-PNU was able to inhibit the growth of T cells from both A24$^+$ and A24$^-$ donors. However, activated A24$^+$ T cells treated with 1 μg mL$^{-1}$ P1C1TM-PNU had notably higher levels of growth inhibition than A24$^-$ T cells (Supplementary Fig. 8), whereas no significant growth inhibition was observed when the T cells were treated with lower concentrations of P1C1TM-PNU.

**In vivo efficacy of P1C1TM-PNU ADC.** To assess the anti-tumour efficacy of the ADC P1C1TM-PNU in vivo, HT29 and HepG2 xenografts were introduced into NSG mice. Mice were given a single dose of 1 or 0.3 mg kg$^{-1}$ of P1C1TM-PNU or a 1 mg kg$^{-1}$ of unconjugated P1C1TM as control 9 days post tumour introduction, when tumour sizes reached ~100 mm$^3$. Single doses of 1 or 0.3 mg kg$^{-1}$ of P1C1TM-PNU or 1 mg kg$^{-1}$ of unconjugated P1C1TM were well tolerated in the treated mice with no observed changes in neither behaviour nor body weight. In mice harbouring HT29 xenografts, treatment with 1 mg kg$^{-1}$ but not 0.3 mg kg$^{-1}$ of P1C1TM-PNU resulted in a 70% inhibition of tumour growth (Fig. 7a). Control of tumour growth was observed even beyond 30 days post treatment. However, in mice harbouring HepG2 xenografts, tumour growths of mice treated with 1 or 0.3 mg kg$^{-1}$ of P1C1TM-PNU were identical to the control group treated with 1 mg kg$^{-1}$ of unconjugated P1C1TM antibodies (Fig. 7a). Hence, the anti-tumour efficacy of

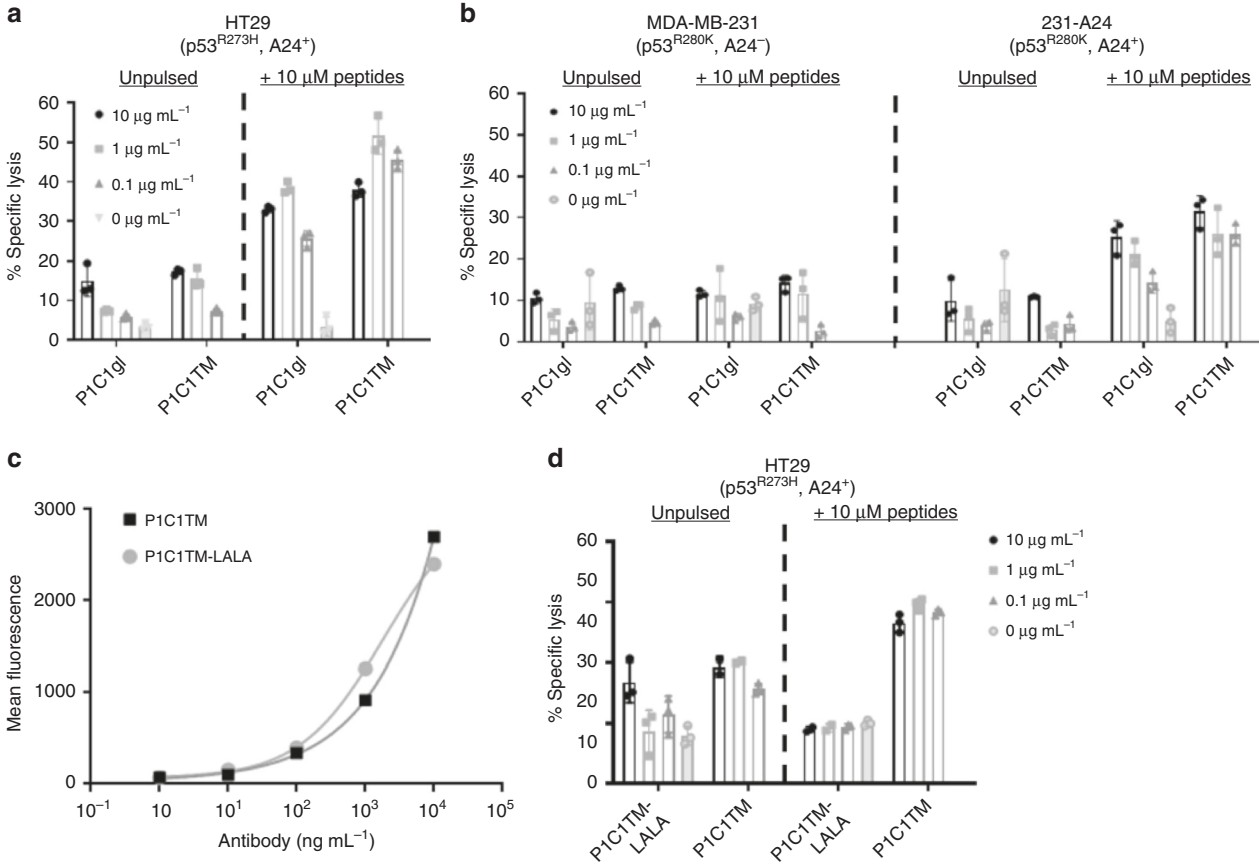

**Fig. 5** P1C1TM mediates ADCC of target cells presenting high levels of p53$_{125-134}$/A24 pMHCs. The ability of P1C1TM to mediate ADCC was assessed using **a** mutant p53-expressing HLA-A24$^+$ colon carcinoma HT29 cells and **b** untransduced or HLA-A24-transduced MDA-MB-231. Target cells were incubated with a range of concentrations of P1C1TM and human PBMCs at an effector:target ratio of 15:1 and in the presence or absence of 10 μM exogenous p53$_{125-134}$ peptides for 16 h. Cytotoxicity was determined by an LDH release assay. Comparison of P1C1TM and the mutant P1C1TM-LALA in their ability to **c** bind the target p53$_{125-134}$/A24 pMHC and **d** mediate ADCC of HT29 cells was done. While the binding avidity between P1C1TM with wild-type Fc and the P1C1TM-LALA mutant with reduced binding to Fc receptor was identical, ADCC was significantly reduced with the P1C1TM-LALA. Data are mean of triplicates ± SEM. ADCC antibody-dependent cellular cytotoxicity.

P1C1TM-PNU was specific for cells expressing mutant p53 and presenting high levels of p53$_{125-134}$/A24 pMHCs both in vitro and in vivo. In a separate experiment, mice bearing HT29 xenografts were treated with an initial dose of 1 mg kg$^{-1}$ followed by two subsequent weekly doses of 0.3 mg kg$^{-1}$. However, while inhibition of tumour growth was observed, mice exhibited reductions in fitness after the third dose of ADCs and eventually succumbed to the treatment (Fig. 7b).

Lastly, we studied the efficacy of P1C1TM-PNU in a pulmonary HT29 metastatic model. HT29 cells were introduced into NSG mice by intravenous tail injection and treated with either 1 mg kg$^{-1}$ P1C1TM-PNU or 1 mg kg$^{-1}$ P1C1TM 24 h after introduction of tumour cells. Twenty days post tumour introduction, mice treated with P1C1TM only lost significant body weight. By day 40, 40% of the control-treated mice had to be sacrificed. On the other hand, mice treated with P1C1TM-PNU had delayed onset of disease and 80% of the mice were able to survive 60 days post tumour introduction before succumbing to disease (Fig. 7c).

**In vivo evaluation of off-target toxicity of P1C1TM-PNU.** While P1C1TM-PNU showed specific cytotoxicity only to HT29 xenografts presenting high levels of p53$_{125-134}$/A24 pMHCs in vivo, its potential in vivo off-target toxicity to healthy tissues had to be addressed. Hence, we treated HLA-A24 transgenic mice

with P1C1TM-PNU with a single therapeutic dose of 1 mg kg$^{-1}$ or left untreated. No evidence of toxicity was observed as judged by their behaviour and weight changes. Pathological evaluation of major organs was then performed 20 days after treatment (Supplementary Table 1). Most organs, including the heart, kidney and bone marrow, exhibited no evident changes in histology (Supplementary Fig. 9). Overall analysis by the pathologist concluded no significant cytotoxicity posed by P1C1TM-PNU.

## Discussion

The selection of a suitable antigen is crucial to the safety and efficacy of any targeted immunotherapy. The ideal tumour-associated antigen (TAAs) would be one that differentiates tumours from normal healthy tissues with great specificity. Unfortunately, a great number of such TAAs are intracellular proteins that are not targetable by conventional antibodies[31,32]. However, the continuous degradation of such proteins into peptides and their presentation as peptide-MHC complexes, recognized by TCRs expressed by CD8$^+$ T cells, are attractive targets for targeted therapeutics with TCRL specificity.

The prevalence of mutations in the *p53* gene in the majority of cancers makes targeting p53 an approach with potentially broad applicability[33,34]. Furthermore, missense mutations resulting in the accumulation of the p53 protein in tumours but not in healthy cells can lead to higher levels of p53-derived pMHC

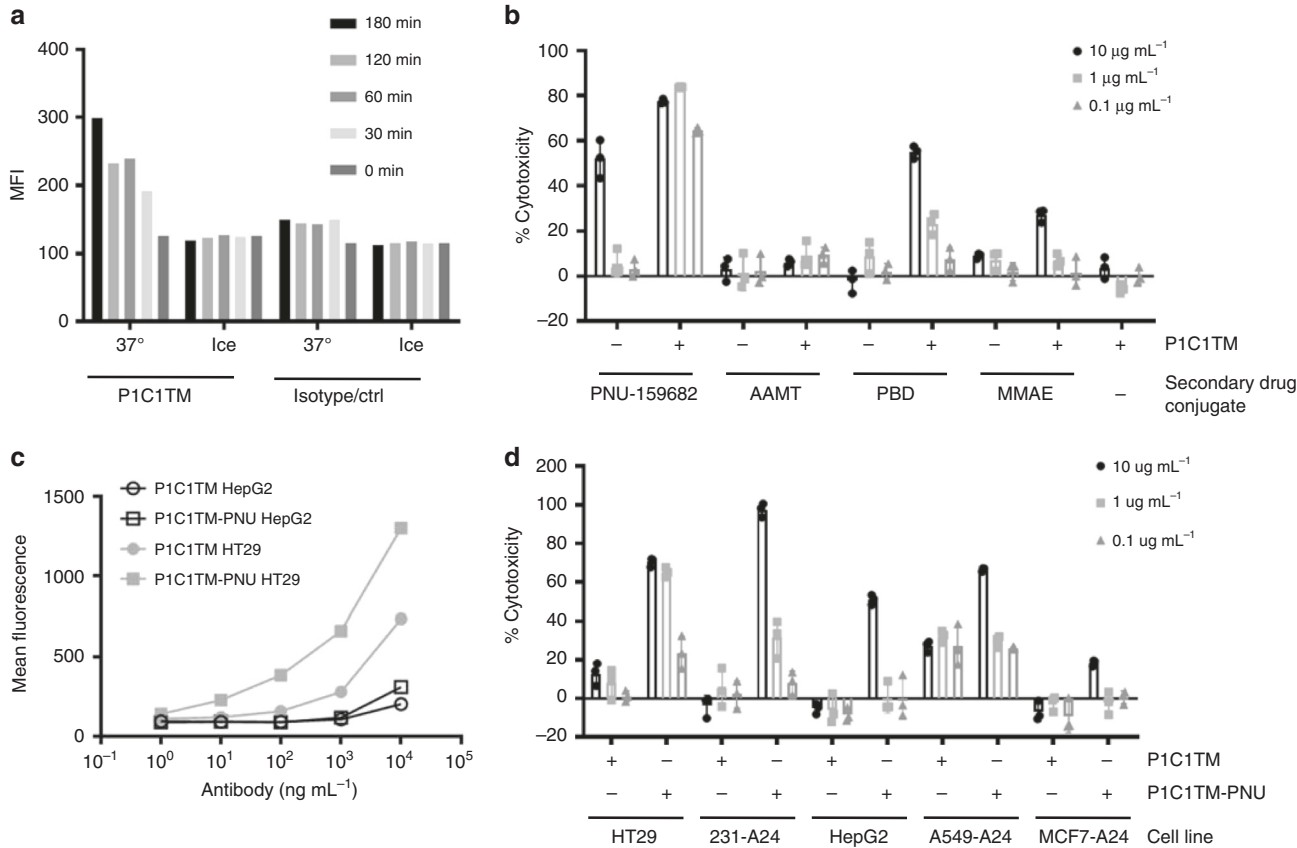

**Fig. 6** Antibody–drug conjugate P1C1TM-PNU mediates cytotoxicity of tumour cells presenting p53$_{125–134}$/A24 pMHC. **a** P1C1TM and an isotype control human IgG1 was conjugated with the pH-dependent dye, pHrodo Red and incubated with HT29 cells at 37 °C or on ice. At different time points, cells were washed with cold PBS and internalization of pHrodo Red-labelled antibodies was assessed by flow cytometry. Data are representative of two independent experiments. **b** HT29 cells were incubated with varying concentrations of P1C1TM in the presence or absence of four different anti-human IgG secondary antibody–drug conjugates in equimolar ratios. Cell viability was assessed after 3 days by an MTS assay. Data are means of triplicates ± SEM. **c** Cytotoxic drug PNU-159682 was directly conjugated to P1C1TM (P1C1TM-PNU) and assessed for binding to HT29 and HepG2 cells by flow. **d** HLA-A24 expressing cell lines expressing mutant p53 (HT29, MDA-MB-231-A24) and wild-type p53 (HepG2, A549-A24, MCF7-A24) were incubated with varying concentrations of P1C1TM only or the P1C1TM-PNU antibody–drug conjugate. Cell viability was assessed by MTS assay after 3 days. Data are means of triplicates ± SEM.

complexes on the surface of the tumour cells. While the MHC complexes may present p53-derived peptides harbouring mutant sequences, these are rare due to strict MHC class I restrictions. However, several immunogenic wild-type p53 peptides that are present in both wild-type and mutant p53 proteins have been reported[17,18,35,36], leading to the development of therapeutic strategies targeting these wild-type p53 pMHCs, including peptide vaccines, soluble TCRs, and more recently a murine TCR mimic antibody, T1-116C, specific for the wild-type p53$_{65–73}$/HLA-A2 pMHC[8,37,38]. Although soluble TCRs have shown therapeutic potential and entered clinical trial, they are innately of low affinity, are challenging to produce and lack cytotoxic functions. Monoclonal antibodies, on the other hand, have well-characterized physical properties, easily produced in large quantities, and naked antibodies possess the innate ability to inhibit tumour cell growth. Hence there is significant interest in the development of antibodies with TCRL specificities specific for pMHC targets, with several groups demonstrating promising therapeutic potential in several cancer models[7–9,11,39,40].

We describe here the isolation and engineering of a fully human antibody, P1C1, with TCRL specificity for the p53$_{125–134}$/HLA-A24 pMHC. Using a panel of cell lines, we demonstrated that binding of the affinity-matured P1C1TM antibody was observed only in the presence of both p53 and HLA-A24

expression. Li et al.[8] previously reported an antibody, T1-116C, specific for the p53$_{65–73}$/HLA-A2 pMHC, in a study involving a panel of cell lines with various p53 statuses. Antibody staining was observed in cell lines expressing either wild-type or mutant p53. Hence, there was no correlation between the level of binding of T1-116C and the mutation status of p53, as significant staining was observed in cell lines expressing wild-type p53 and even on HL-60 cells that do not express p53 or HLA-A2. In our hands, we observed that P1C1TM binding was strong on cell lines expressing mutant p53, but only minimal on those expressing wild-type p53. Importantly, no staining was observed in cells that expressed HLA-A24 but not p53.

The degradation of wild-type p53 through the ubiquitin proteolysis pathway is dependent on its interaction with the E3 ligase, MDM2. Mutations in p53 have been classified as contact mutants, in which mutations affect the DNA-binding function of p53 without affecting the protein integrity, or structural mutants whereby the mutation results in unfolding or aggregation of the protein[41]. The structure of the p53 molecule affects the way it interacts with MDM2[42], hence while MDM2 is still able to interact with certain mutant p53 molecules, it fails to mediate the degradation of these mutant proteins[43], leading to the accumulation of mutant p53 proteins. HT29 cells harbour the R273H contact mutation that leads to reduced DNA binding without

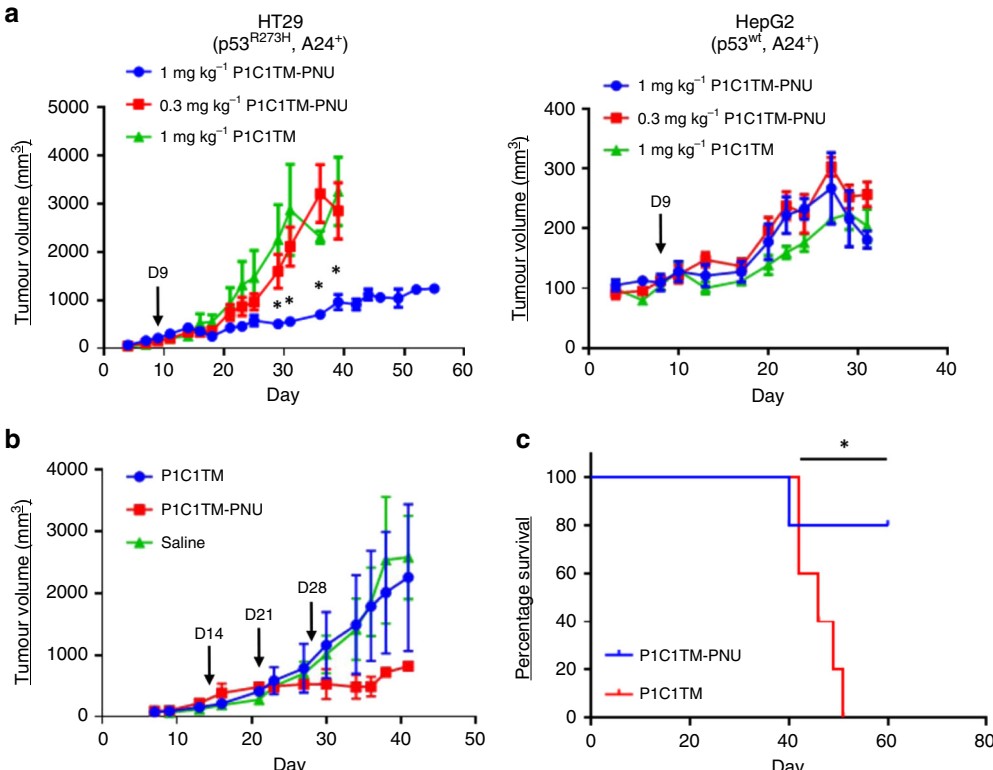

**Fig. 7** P1C1M-PNU limits growth of mutant p53-expressing HT29 xenograft in vivo. HLA-A24 expressing **a** HT29 (p53[R273H]) and HepG2 (p53[wt]) cells were injected subcutaneously into NSG mice. After tumours reached an average size of ≥100 mm³, mice were treated with a single dose of either P1C1TM-PNU ADC (1 or 0.3 mg kg⁻¹) or unconjugated P1C1TM only (1 mg kg⁻¹) as indicated by the arrows. P1C1TM-PNU significantly delayed tumour growth as compared to the unconjugated P1C1TM (* represents significant differences with $p \leq 0.05$. Statistical analyses were done by Student's $t$ test with Holm–Šidák method of multiple comparison.). **b** Mice bearing HT29 (p53[R273H]) tumours were treated with a loading dose of 1 mg kg⁻¹ of either P1C1TM-PNU or unconjugated P1C1TM only, followed by two more doses at 0.3 mg kg⁻¹ 7 days apart. **c** HT29 cells were introduced i.v. into NSG mice and treated with 1 mg kg⁻¹ P1C1TM-PNU or unconjugated P1C1TM 24 h post introduction. P1C1TM-PNU-treated mice showed significantly prolonged survival compared to unconjugated P1C1TM-treated mice (*$p = 0.0339$. Statistical analysis was done with a log-rank test for survival.). Tumour sizes were calculated as $V = \frac{1}{2}(\text{length} \times \text{width}^2)$. Data are means of triplicates ± SEM.

compromising the p53 protein structure[44,45]. In addition, it has been shown that the mutant p53 in HT29 cells continues to be associated with Mdm2 albeit at lower affinity[43]. Here we show that p53[125–134]/HLA-A24 complexes were readily detected on the surface of HT29 cells by P1C1TM. Treatment of HT29 cells with the MDM2 inhibitor, nutlin, did not result in any change in the level of intracellular p53 or surface p53 peptide/MHC complex. This is in agreement with previously reported observation that while the mutant p53 in HT29 cells can associate with MDM2, the E3 ligase had no effect on p53 stability[43]. However, when HT29 cells were treated with interferon-γ, surface levels of HLA-A24 increased along with a similar increase in P1C1TM staining. Also, treatment of HT29 cells with the proteasome inhibitor MG132 yielded a decrease in surface levels of HLA-A24 and a concomitant decrease in P1C1TM staining, as a result of the reduction in proteasomal activity required to generate peptides that stabilize the pMHC complexes on the surface of the cells. Thus, collectively these data show that while mutant p53 accumulate to high levels intracellularly in tumour cells, continuous degradation through MDM2-independent, proteasome-dependent pathways generates peptides that are eventually presented on the cell surface as pMHC complexes and are attractive targets for immunotherapy.

The specificity of the TCRL antibodies for the peptide presented is critical for the safety and efficacy of TCRL antibody-based therapeutics. Unlike TCRs, an antibody's specificity is not naturally restricted to pMHC complexes. Structural studies and comparisons by various groups have observed that antibodies with TCRL specificity do not necessarily recognize and bind the target pMHC in the conserved TCRL orientation[46–48]. The orientation and the fine specificity of the antibody for the pMHC will determine the level of off-target cross-reactivity[46]. Thus, it is of great importance that we validated P1C1TM's fine specificity for the p53[125–134]/HLA-A24 pMHC. We showed that P1C1TM interacted effectively with the central residues of the p53[125–134] peptide and exhibited almost undetectable cross-reactivity to numerous peptides with homologous sequences. Interestingly, P1C1TM does not bind the murine p53[125–134] peptide that differs only at positions 5 and 9. Structural studies are underway to elucidate the molecular interactions and fine specificity between P1C1TM and the p53[125–134]/HLA-A24 pMHC.

Antibodies with TCRL specificity have been observed to have anti-tumour activity in vitro and in vivo[7,8,11,13–15]. Similarly, we demonstrated that P1C1TM was able to mediate ADCC of mutant p53-expressing cells presenting high levels of p53[125–134]/HLA-A24 complexes. Alternatively, antibodies can be armed with cytotoxic payloads, such as DNA-damaging drugs and inhibitors of tubulin polymerization, for delivery to tumour cells, resulting in their cytotoxicity. The kinetics of internalization and expression levels of the target antigen are two important points for consideration for the development of such ADCs[49]. Constitutive internalization of pMHCs by a clathrin- and dynamin-independent mechanism have been reported[50], favouring the development of TCRL ADCs. Hence, we explored the possibility

of targeting mutant p53-expressing tumours with a p53-derived pMHC-specific TCRL ADC. However, tumour-specific pMHCs are low in density on the surface of target cells as compared to the current ADC targets such as HER2. A comparative efficacy study between two separate ADCs, based on the same anti-Her2 antibody trastuzumab, showed that the selection of appropriate drugs may overcome the low antigen density[51]. A recent study also showed that in addition to direct anti-tumour effects, the anti-HER2 targeting ADC was able to enhance anti-tumour immune responses through the activation of antigen-presenting cells and also increasing MHC class I expression on tumour cells[52]. Such an adjuvant effect could potentiate the efficacy of TCRL ADCs. Lowe et al.[53] and others have demonstrated that TCRL ADCs conjugated with duocarmycin and immunotoxins were capable of mediating cytotoxicity in tumour cells expressing low levels of target pMHC complexes[9,10,40].

Consistent with these studies, we observed the ability of the $p53_{125-134}$/HLA-A24-specific P1C1TM antibody to specifically deliver the DNA-targeted cytotoxic drugs PDB and PNU-159682, to inhibit mutant p53$^+$ tumour cell growth in vitro. Importantly, the cytotoxic effects of P1C1TM-PNU ADCs were restricted to only mutant p53-expressing colorectal cancer cells in in vivo models of colon adenocarcinoma. Cytotoxicity was also observed in activated T cells treated with high concentrations of P1C1TM-PNU. This may be due to differences in sensitivity to the PNU-159682 as Jurkat T cells were found to be >5-fold more sensitive to PNU-159682 than to HT29 cells[54]. Resting T cells do not present significant levels of $p53_{125-134}$/HLA-A24 complexes and thus may more accurately represent healthy tissues in vivo. Furthermore, our study with HLA-A24 transgenic mice, where all tissues potentially present a myriad of HLA-A24-restricted pMHCs, showed that the therapeutically effective dose was well tolerated with minimal pathology observed. While the HLA-A24 antigen repertoires presented by human and mouse are different, this study nevertheless provided further specificity evidence at the proteome scale. Hence, in our proof-of-concept studies described, while therapeutic effects were modest, we observed evidences of specificity and safety of targeting p53-derived pMHCs with a TCRL ADC. Further improvements in the efficacy and safety of the drug conjugate can be achieved with better selection of linker, payload and conjugation chemistry[49,55].

Over-expression of the p53 protein in tumour cells allows p53-derived pMHCs to be potential targets for immunotherapy[17,18,35,56–59]. However, the specificity of immunotherapeutic strategies against such antigens is highly dependent on the threshold level of antigens required for binding and activation[20,32,60]. Tumour-associated pMHCs are relatively low in density, but are known to be arranged as clusters on the cell surface[61], facilitating the binding of multivalent molecules such as antibodies. We speculate that the ability of our P1C1TM to discern the subtle differences in antigen levels may be attributed to its moderate rather than high affinity. This has been proven to be crucial in the reduction of on-target/off-tumour non-specificity, especially in chimeric antigen receptor (CAR) T cells[20,62–64]. In a recent study by Liu et al.[65], affinity-tuned anti-HER2 CAR-T cells demonstrated dramatically different efficacy and toxicity profiles against tumour and normal tissues. Similarly, P1C1TM may strike such affinity balance to achieve discrimination between cells expressing wild-type and mutant p53, presenting a promising therapeutic strategy for a wide range of cancers.

## Methods

**Cell lines and reagents**. MDA-MB-231 (expressing p53 with a mutation at Arg280 to a Lys, i.e. p53$^{R280K}$) (cat. #HTB-26), A549 (wild-type p53 expressing, i.e. p53$^{wt}$) (cat. #CCL-185), BT474 (p53$^{R280K}$) (cat. #HTB-20), MCF7 (p53$^{wt}$) (cat. #HTB-22) and HepG2 cells (p53$^{wt}$) (cat. #HB-8065) were purchased from ATCC.

Cells were cultured in either RPMI-1640 or Dulbecco's modified Eagle's medium (DMEM) supplemented with 10% foetal bovine serum and 1% penicillin/streptomycin, in humidified CO$_2$ (5%) incubator at 37 °C. The HLA*A24:02 heavy chain was cloned into the pLVX-IRES-Zsgreen1 lentiviral expression vector and lentivirus particles were produced according to the manufacturer's protocol (Clontech Laboratories Inc). MDA-MB-231, A549, BT474 and MCF7 cells were then transduced, and the successfully transduced cells were selected by fluorescence-activated cell sorting for Zsgreen1 green fluorescent protein expression. SaoS2 (biallelic deletion of TP53 gene, i.e. p53$^{null}$) and HT29 (expressing p53 with a mutation at position Arg273 to His, i.e. p53$^{R273H}$) cells were a kind gift from Dr. Wang Bei (SIgN, A*STAR) and Dr. Lee-Ann Hwang (p53Lab, A*STAR), respectively. The cells were cultured as described above in DMEM and McCoy, respectively. Nutlin and MG132 were kind gifts from Prof. Sir David Lane (p53Lab, A*STAR), and recombinant interferon-γ was purchased from Miltenyi Biotec. Cell lines are routinely checked for mycoplasma contamination.

The p53 statuses of cell lines used can be found on the TP53 database (http://p53.fr).

**Ethics**. All experiments involving animals were performed in accordance with guidelines approved by the Institutional Animal Care and Use Committee of the Biological Resource Centre (BRC), Agency for Science, Technology and Research (A*STAR), Singapore. Use of the apheresis blood from healthy donors for this study was approved by the National University of Singapore Institutional Review Board (IRB reference number: 2017/2512).

**Peptides**. All peptides were synthesized by Genscript USA Inc. at >90% purity and dissolved in dimethyl sulphoxide at 40 mg mL$^{-1}$ and frozen at −80 °C. The peptides hTERT$_{324–332}$ and hTERT$_{461–469}$, WT1$_{235–243}$, WT1$_{417–425}$, p53$_{125–134}$ and p53$_{204–212}$ were all previously described by various groups as HLA*A24:02-binding peptides[17,56,66–68]. Biotinylated and non-biotinylated p53$_{125–134}$/A24 pMHC and control peptide-MHCs were synthesized either by UV-peptide exchange or refolded with recombinant HLA-A24 heavy chain and β2-microglobulin[69]. Briefly, peptide exchange was performed by exposing 2.1 μM recombinant HLA-A24 conditional peptide-MHC to 15 min UV irradiation in the presence of 50 μM peptides in phosphate-buffered saline (PBS) on ice, followed by an hour incubation at 37 °C. Refolding of recombinant peptide-HLA-A24 complexes was done in vitro by the rapid dilution of protein inclusion bodies of the α-chain of HLA-A24 and β2-microglobulin in the presence of peptides. Refolded complexes were then concentrated, dialysed, biotinylated and lastly purified by size-exclusion chromatography on an Äkta Pure Fast Protein Liquid Chromatography system (GE Healthcare).

**Biopanning for p53$_{125–134}$/A24-specific TCRL antibodies**. Biopanning of a naive human Fab phage display library (Humanyx Pte Ltd)[19] was carried out to isolate p53$_{125–134}$/A24-specific TCRL antibodies. Briefly, the phage library is first subjected to negative selection with uncoated M280 streptavidin magnetic beads (Life Technologies) before incubating with soluble biotinylated p53$_{125–134}$/A24 pMHCs. The concentration of p53$_{125–134}$/A24 pMHC used was 100 nM in the first round, 50 nM in the second round and 10 nM in the third round. The input phage in the first round was 10$^{13}$ and 10$^{11}$ CFU for the subsequent rounds. Bead-bound phages were washed with washing buffer (PBS with 0.1% Tween-20) with increasing stringency from 4 times to 20 times from round 1 to round 3. Phage eluted with 100 mM triethylamine was used to infect TG1 E. coli cells (Lucigen) and rescued with M13KO7 helper phage (New England Biolabs). After three rounds of biopanning, Fabs of selected clones were expressed in HB2151 E. coli cells for screening by ELISA. Unique p53$_{125–134}$/A24 pMHC-binding clones were subsequently identified by DNA fingerprinting with BstNI restriction digest (New England Biolabs) and DNA-sequencing.

**Enzyme-linked immunosorbent assay**. Screening of p53$_{125–134}$/A24-specific antibody leads was performed by sandwich ELISA. The 96-well plates were pre-coated with 5 μg mL$^{-1}$ NeutrAvidin (Thermo Fisher Scientific, cat. #31000) in 50 mM carbonate buffer pH 9.6. Fab or IgG bound to NeutrAvidin-captured biotinylated peptide-MHC was detected by incubation with horseradish peroxidase-conjugated anti-human IgG F(ab')$_2$ (cat. #109-036-097) or anti-human IgG Fcγ secondary antibody (cat. #109-035-098) (Jackson Immunoresearch, both 1 in 3000 dilution), respectively. The colorimetric signal observed upon incubation with 3, 3′, 5, 5′-tetramethylbenzidine (SurModics) substrate was quenched with 1 M HCl and absorbance (Abs) was quantified at 450 nm on the EnSpire microplate reader (Perkin Elmer).

Binding affinities of p53$_{125–134}$/A24-specific antibodies were characterized by ELISA as described above, but with wells coated with antibodies. Subsequently, biotinylated peptide-MHCs that bind to the coated IgG were detected with horseradish peroxidase (HRP)-conjugated streptavidin (BioLegend, cat. #405210, 1 in 3000 dilution).

Stability of the HLA-A24 complexes was determined by a stability ELISA[69]. Briefly a 96-well half-area plate (Corning) was coated with 2 μg mL$^{-1}$ NeutrAvidin (Thermo Fisher Scientific) and used to capture 1.6 nM biotinylated peptide-exchanged HLA-A24 pMHC. Peptide-rescued HLA-A24 pMHC complexes were

then probed with 1 µg mL$^{-1}$ anti-β2-microglobulin antibody (Clone TÜ99, BD Pharmingen), followed by HRP-conjugated anti-mouse IgG secondary antibody (Jackson Immunoresearch, 1 in 3000 dilution).

**IgG expression and purification**. p53$_{125-134}$/A24-specific antibodies were reformatted from Fabs into human IgG1 by cloning into the pTT5 vector and expressed in HEK293-6E cells as previously described by Durocher et al[70]. Construction of the triple mutant P1C1 and P1C1TM-LALA were done by combining mutations from individual clones into the P1C1 wild-type sequence using the Quikchange Lightning Multi Site-directed Mutagenesis Kit (Stratagene). Both the vector and cells were obtained from National Research Council of Canada. Antibodies were purified from the culture supernatant using Protein G resin (Merck Millipore).

**Flow cytometry**. Binding avidity and specificity of p53$_{125-134}$/A24-specific antibodies were analysed by pulsing SaoS2 cells in the presence of wild-type p53 peptide or a panel of A24-restricted peptides for 1 h at 37 °C prior to staining. Pulsed cells were then washed with staining buffer (1% bovine serum albumin (BSA) in PBS) to remove excess peptides followed by staining with p53$_{125-134}$/A24-specific antibodies. Bound antibodies were detected by incubation with 1 µg mL$^{-1}$ polyclonal Alexa Fluor 647-conjugated goat anti-human IgG secondary antibodies (anti-human IgG-AF647) (Life Technologies, cat. #A-21445). Staining of endogenous p53$_{125-134}$/A24 was done with the HLA-A24+ cell lines HT29, HepG2 and SaoS2, and the HLA-A24-transduced and -untransduced cell lines A549, MDA-MB-231, BT474 and MCF7. PBMCs were isolated from buffy coats obtained from the Blood Bank of Health Sciences Authority (Singapore), using Ficoll-Hypaque density gradient centrifugation. T cells were purified from PBMCs by magnetic separation using a Pan T Cell Isolation Kit (Miltenyi Biotec), and activation was done with T cell TransAct (Miltenyi Biotec) according to the manufacturer's instructions. Expression of HLA-A24 on T cells was assessed using an Alexa Fluor 647-conjugated anti-HLA-A24 antibody (clone 17A10, MBL International, cat. #K0208-A64, 1 in 1000 dilution). Gating strategy is provided in Supplementary Fig. 10.

Intracellular p53 staining was done with DO-1 (cat. #MA5-12571) or DO-7 (cat. #MA5-12557) antibodies followed by an r-phycoerythrin-conjugated anti-mouse IgG secondary antibody (Life Technologies, cat. #P-852). Cells were first fixed and permeabilised using the Fixation/Permeabilization solution (BD Biosciences) and stained with 5 µg mL$^{-1}$ of primary antibodies followed by 1 µg mL$^{-1}$ of secondary antibodies.

For internalization assays, antibodies were directly conjugated with the pHrodo® Red Microscale Labelling Kit (Thermo Fisher) according to the manufacturer's instructions. HT29 cells were incubated with labelled antibodies at 37 °C or on ice. At indicated time points, cells were transferred to ice-cold staining buffer (1% BSA in PBS) and analysed by flow cytometry. All data were collected on FACSVerse™ (Becton Dickinson) and analysed with the FlowJo Software.

**Affinity maturation**. Affinity maturation of the selected clone P1C1 was performed by phage display using a library in which restricted CDR diversity was introduced[21]. Briefly, the library was constructed such that the CDRs of both the heavy and light chains were randomized with a degenerate codon that encoded 4 main amino acids (tyrosine, alanine, aspartate and serine). Biopanning was performed as previously described with increased stringency by decreasing the binding p53 peptide-MHC concentration from 20 to 1 and 0.2 nM in the first, second and third round of panning, respectively. Additionally, to remove rapidly dissociating clones, the bead-bound phages were subjected to dissociation in increasing number of washes and temperature of wash buffer with each sequential round of panning. Positive clones were identified by ELISA as previously described, and DNA sequenced to further isolate unique and dominant clones. Off-rate comparison of these dominant clones was performed by bio-layer interferometry using BLITZ (ForteBIO) to select clones with improved binding for p53$_{125-134}$/A24. Briefly, biotinylated pMHCs were pre-immobilized onto streptavidin biosensors and their binding to mutant Fab clones were compared to the parental P1C1 clone.

**Off-targeting binding analysis of P1C1TM**. Possible cross-reactive peptides were identified by a motif search of the KEGG database with the search criteria XYSXXLXKXF, XYSXXLXKXF and XYXXAXXKXF. HLA-A24 binding affinities of the short-listed peptides were predicted using NetMHC 3.0 and the 14 top binding peptides along with the murine p53$_{119-128}$ peptide (TYSPPLNKLF) were synthesized for further cross-reactivity analysis.

**Surface plasmon resonance analysis**. Antibody–protein interaction was performed on the Biacore™ T200 (GE Healthcare Life Science). Briefly, a mouse anti-human IgG (CH2 domain) (Thermo Fisher, clone R10Z8E9, cat. #MA5-16929) antibody was immobilized on a CM5 sensor chip (GE Healthcare Life Science) by amine coupling chemistry. Anti-p53$_{125-134}$/A24 antibodies were captured by the immobilized anti-human IgG antibodies and subsequently binding kinetics were assessed using six concentrations of p53$_{125-134}$/A24 monomers. Antibodies and pMHCs were diluted in HBS-EP (0.01 M HEPES, pH 7.1, 0.15 M NaCl, 3 mM EDTA, 0.005% (v/v) Surfactant P20, GE Healthcare Life Sciences). All measurements were done at 25 °C. The sensorgrams were referenced against a control lane

(no immobilized anti-human IgG). Rate constants and affinity were determined using the BIAevaluation software and fitted to a 1:1 Langmuir binding model. The $\chi^2$ values of the curve fits are indicated in the respective figures.

**ADCC assay**. PBMCs were isolated from buffy coats obtained from the Blood Bank of Health Sciences Authority (Singapore), using Ficoll-Hypaque density gradient centrifugation. Target cells HT29, MDA-MB-231 and 231-A24 were pulsed or unpulsed with p53$_{125-134}$ peptides and incubated with TCRL antibodies at various concentrations and PBMCs at an effector/target ratio of 15:1 for 16 h. Cytotoxicity was then measured by lactate dehydrogenase (LDH) release in the supernatants with CytoTox 96 Non-Radioactive Cytotoxicity Assay Kit (Promega) according to the manufacturer's instructions.

**ADC cytotoxicity assay**. Anti-human secondary antibody–drug conjugated with the cytotoxic drugs MMAE (anti-HuFc-MMAE, cat. #AH-102AE), PNU-159682 (anti-HuFc-PNU-159682, cat. #AH-102PN), pyrrolobenzodiazepine (anti-HuFC-PBD, cat. #AH-106PB) and AAMT (Fab-anti-HuFC-AAMT, cat. #AH-205AM) were purchased from Moradec. P1C1TM was conjugated to PNU-159682 via a vc-PAB linker, to a DAR of about 4.1 (Levena Biopharma). For indirect killing assays, target cells (20,000 per well) were first incubated with P1C1TM at various concentrations for 30 min before the addition of the secondary antibodies at equimolar concentrations. For direct killing assays, target cells (20,000 per well) were incubated with P1C1TM-PNU at various concentrations. Assays were incubated for 3 days at 37 °C and viable cells were quantified with the either the CellTiter 96© AQueous One Solution Cell Proliferation Kit (Promega) or the alamarBlue Cell Viability assay (Thermo Fisher Scientific) according to the manufacturer's instructions. Abs was measured at 490 nM using a microplate reader (EnSpire™, Perkin Elmer). Fluorescence was read at an excitation wavelength of 560 nm and an emission wavelength of 590 nm. Viability was calculated as Abs$_{treated}$/Abs$_{untreated}$ × 100% or Fl$_{treated}$/Fl$_{untreated}$ × 100%. Cytotoxicity was calculated as % cytotoxicity = (100% − %viability).

**Animals**. NSG mice were purchased from InVivos Pte Ltd and bred under specific pathogen-free conditions. HLA-A24 transgenic mice were purchased from Taconic Biosciences. These mice carry a transgene consisting fragments of the human *HLA-A*24:02* gene and mouse *H2-K$^b$* gene, which encodes a chimeric class I molecule consisting of the human HLA-A24 leader, α1 and α2 domains ligated to the murine α3, transmembrane and cytoplasmic H2-K$^b$ domains.

**In vivo specificity of p53/A24-specific TCRL antibody**. Six- to eight-week-old NSG mice were subcutaneously implanted with $2.5 \times 10^6$ HT29 in the lower right flank and $5 \times 10^6$ MDA-MB-231 (or $2.5 \times 10^6$ SaoS2) in the lower left flank. Tumour volume was determined using an external caliper measuring the greatest longitudinal diameter (length) and the greatest transverse diameter (width). Tumour volume ($V$) was calculated using the modified ellipsoid formula $V = ½$ (length × width$^2$). Antibodies used for imaging were conjugated with Alexa Fluor 680 using Invitrogen's SAIVI Rapid Antibody Labelling Kit. Upon establishment of tumours (≥100 mm$^3$), 50 µg of Alexa Fluor 680-labelled P1C1TM IgG was intravenously injected through the tail vein of each mouse. Binding of the labelled antibody to tumours was detected using the IVIS® Imaging System (Perkin Elmer) 48 and 120 h post antibody administration and quantified by the Living Image 3.2 software.

**In vivo efficacy of p53/A24-specific TCRL ADC**. Six- to eight-week-old NSG mice were subcutaneously implanted with $1.5 \times 10^6$ HT29 or $2 \times 10^6$ HepG2 in the lower right flank. Tumour volume was determined as described above. Upon establishment of tumours (≥100 mm$^3$), mice were randomly assigned into treatment and control groups ($n = 5$) and treated with single or multiple doses of 1 or 0.3 mg kg$^{-1}$ P1C1TM-PNU intravenously. Mice in the control group were treated with similar doses of the unconjugated P1C1TM antibody. Tumour volume was measured thrice weekly for 30 days post treatment or as described.

For the metastatic model, $1.0 \times 10^6$ HT29 cells were injected intravenously into the NSG mice. At 24 h after introduction of tumour cells, the mice were treated with a single dose of 1 mg kg$^{-1}$ P1C1TM-PNU or P1C1TM antibody only ($n = 5$ per group). Body weight and survival was monitored, and mice were sacrificed when a drop of more than 20% in body weight was observed.

**In vivo cross-reactivity and toxicity of p53/A24-specific TCRL antibody**. Six- to eight-week-old HLA-A24 transgenic mice were intravenously injected a single dose of P1C1TM-PNU or P1C1TM at 1 mg kg$^{-1}$ ($n = 5$ per group). Body weight and fitness of mice were monitored thrice weekly. Mice were sacrificed 20 days post tumour injection for histopathological studies (Advanced Molecular Pathology Laboratory, A*STAR).

**Statistical analysis**. All data were compiled and analysed by GraphPad Prism 6.0. Data presented are means ± SEM and non-linear regression analyses were used to fit curves. Flow cytometry binding data are representative of three or more independent experiments. Due to variability in PBMC donors, ADCC assays were not

statistically comparable, but data presented are means of triplicates in individual experiments representative of three or more independent experiments. Statistical analyses of animal experiments were done by Student's $t$ test with Holm–Šidák method of multiple testing correction and statistical significance was defined with $\alpha$ <0.05. Statistical analysis of the animals treated in the metastatic model was done by a log-rank (Mantel–Cox) test.

**Reporting summary.** Further information on research design is available in the Nature Research Reporting Summary linked to this article.

### Data availability

The source data underlying Figs. 1–3, 5–7 and Supplementary Figs. 3, 4, 6 and 8 are provided as a Source Data file. All the other data supporting the findings of this study are available within the article and its supplementary information files and from the corresponding author upon reasonable request. A reporting summary for this article is available as a Supplementary Information file. P1C1 and the affinity-matured mutants are proprietary and can only be obtained through a Materials Transfer Agreement.

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

## Acknowledgements

We thank Evan Newell for providing UV-cleavable peptide/HLA-A24 complex and materials for peptide-MHC production; John E. Connolly and David P. Lane for their helpful suggestions; Chee Beng Ong for the insightful discussion and the Advanced Molecular Pathology Lab (AMPL) at the Institute of Molecular and Cell Biology, A*STAR for their services in the pathology evaluations. We also thank Dr. Anis Larbi and the Singapore Immunology Network flow cytometry platform for their assistance and advice. This work is supported by an Industrial Alignment Fund (IAF311007) from Economic Development Board (EDB) and Biomedical Research Council (BMRC), Agency for Science, Technology and Research (A*STAR), Singapore. The flow cytometry platform is part of the SIgN Immunomonitoring platform and is supported by an Industrial Alignment Fund (IAF311006) from the BMRC, A*STAR and BMRC transition funds (#H16/99/b0/011).

## Author contributions

L.L., A.G. and C.W. designed the research, analysed the data and wrote the paper. L.L. and A.G. designed and carried out the antibody engineering. L.L., A.G., J.K. and S.L. carried out the in vitro experiments. A.G. carried out the mouse experiments.

## Competing interests

The authors declare no competing interests.
