## [Peer Review File · Nature Communications]

Reviewers' Comments:

Reviewer #1:

Remarks to the Author:

Low et al. report the identification, isolation, engineering, biochemical and functional characterization of a T cell receptor-like (TCRL) monoclonal antibody (mAb) that recognizes the molecular complex of the p53-derived peptide p53125-134 bound to the human MHC-I molecule HLA-A24. Overall, the reagent described is novel, and the experiments well-performed. The characterization of the mAb suggests that this reagent deserves further study as a potential diagnostic or therapeutic agent in humans. Conceptually, the work is not novel, but follows an established approach of screening phage display libraries for Fab that bind a recombinant protein antigen, refining the phage library for identification of higher affinity reagents, and then engineering the Fab into an IgG1 to permit better characterization.

Although the paper is generally written clearly, several points need to be explained a little better in revision, and a number of detailed points that I will enumerate below. One nagging confusion is that the mAb is against the wildtype p53 sequence, and that the p53 mutants that are tested are mutants with respect to the function/expression of p53—thus, as I understand this, the mAb is detecting increased levels of the p53peptide/MHC complex resulting from dysregulation of the expression of the epitope rather than of the generation of a mutant epitope. This needs to be made clear from the start.

Specific points to be addressed in revision:

- p. 1. Make clear that the mAb sees the WT p53 peptide in association with A24.
- p. 3. In the discussion of the mutants somewhere make it clear that the mAbs are not to tumor specific antigens.
- p. 3. Period after oncogenesis.
- p. 3. To remove mutant p53 in tumor cells (unclear).
- p. 4. Degraded and presented as peptide-MHC antigens
- p. 4. Define TCR-like antibody (TCRL) early on, please, not in a section heading.
- p. 6. The description that affinity was low, is never backed up quantitatively.
- p. 6. pMHC antigen on “the” cell surface. Crucial, not pivotal.
- p. 7. Individual clones “were” screened

p. 7. Due to “the” small theoretical diversity.

p. 7. And throughout—what precisely is the HLA-A*24 that is being used here? A*24:02

p. 9. Affinity matured P1C1TM.....

throughout---although the mAb sees A*24 with peptide, there is no definitive experiment that shows that it fails to see the same peptide bound to a different allelic product MHC-I.

p. 13 and throughout...listing the pheno/genotype of the cell lines as p53R273H for instance, fails to emphasize that these are mutations that result in different levels of expression, and possibly different levels of presentation of the wild-type peptide.

p. 19 (BiTEs) target a cancer

p. 19 cells through the anti-CD3 arm

try to use TCRL throughout otherwise use TCR-like.....one or the other.

p. 21 what is the affinity that is called “moderate”

p. 23 The p53 move up to the next line.

p. 25 reference for Durocher et al?

p. 28 equilibrium dissociation constants (KD), kinetic constants k_a and k_d in figure .

also, what are the statistics of the curve fits for the binding parameters?

p. 29 what genotype of HLA-A*24 for the transgenics

p. 30 100 mm³

What is the journal’s policy on MTA dependent provision of materials. Once reported the mab should be available for experimental purposes.

p. 40. P1C1TM and an isotype control.....were conjugated.

p. 44. Figure 2. C. equilibrium constant KD, kinetic constants k_a , and k_d . What are errors for the values?

p. 45. Figure 3. Mean fluorescence or MFI? How many times were these experiments done/

p. 48 is growth inhibition the same as cytotoxicity?

Suppl fig 2 same concerns for values and statistics as above.

Comment on the peptide scan analysis. Since these seem to have been done only at one concentration, one should be careful not to overinterpret the results.

Reviewer #2:

Remarks to the Author:

The article by Wang and collaborators describes the selection of an antibody specific for a p53/125-135 peptide complexed to HLA-A24, which mimics the reactivity of a T cell receptor. The strategy to isolate this antibody from a phage display library, followed by mutational affinity maturation, is well thought-out and well performed. Most of the experiments presented have appropriate controls and show that the affinity matured antibody is able to recognize a non-mutated p53 peptide only in tumor cells that express mutations in other regions of the p53 molecule, presumably because these tumors overexpress p53, or perhaps because the mutated p53 proteins are processed differently. Unfortunately, the authors do not distinguish the 2 possibilities, which could be done using this antibody, for example, by overexpression of WT p53 in an HLA-A24+ cell line.

In addition to the above issue, there are major points that need to be addressed:

- 1) The authors have failed to take into account that activated immune cells, in particular, T cells express elevated wt p53 and are potential targets for any probe targeting this antigen, especially one that has been engineered to be “high affinity” against a non-mutated p53 epitope. Supplemental Table 1 indicates “no significant findings” relating to toxicity of the probe. The toxicity studies in HLA-A24 transgenic mice have many caveats since the mouse p53 sequence in this epitope is different from and non-crossreactive with the human sequence (Fig. 3). At minimum, in vitro based experiments involving mitogen activated HLA-A24+ human T cells as targets are needed.
- 2) The authors assume that because tumors that do not have p53 mutations, they do not react with the antibody, normal cells will also not react with the antibody. However, many tumors express lower levels of MHC-I as compared to normal tissues, so peptide/MHC density could play a role in potential reactivity in normal tissues. Can the antibody be used in histological sections to stain tumors (with and without p53 mutations) and normal tissue samples in HLA-A24 individuals? These experiments could provide additional information regarding potential off-target toxicities.
- 3) In the experiment with results shown in Fig. 7, why did the authors limit treatment to a single dose of antibody? Multiple doses (e.g., weekly) could have provided a better therapeutic effect. Also, experiments with minimal residual disease (treating at an earlier time point) or a “metastatic” model intravenous injection monitoring tumor growth in liver/lungs with bioluminescence would greatly improve the message of therapeutic potential of the antibody.
- 4) Although the use of xenograph-based experiments does demonstrate the efficacy of the probe, such “cutting edge experiments” now require the use of orthotopic xenographs derived from

specimens obtained from patients, which have been characterized relative to potential defects in antigen processing and presentation machinery, which generally are commonly found in human tumors.

5) The manuscript has too much data presented, in both the text figures and supplemental figures; there are too many tumor cell lines and variables involving A24 and p53 status to keep straight. The whole paper should be streamlined. While the panel of tumor cells lines utilized is extensive, the use of any target cell, whether naturally presenting A24 or, more importantly, transfected to express A24 points out that the levels of A24 expression and defects in antigen processing and presentation (no evidence of levels of efficacy of transfection in these cell lines is indicated), need to be understood to relate the specificity of the probe. Moreover, there is 1) no mention of any potential shared HLA specificities among the non-transfected positive and negative tumor cell lines used that also might be capable of presenting the epitope, and 2) no mention of the levels of epitope that are naturally presented by tumors that are recognized by T cells to get a better understanding of the affinity of the probes.

Minor Comments:

1) Many of the figures are not as well defined as they could be. The use of APC to denote the target cells (Figure 1) rather than the actual tumor cell is confusing. In many figures the descriptions of the cell lines vary from one figure to the next and/or the text. Sometimes the p53 mutational site is given, other times it "p53 mut", "p53 pos" or p53.

2) The authors fail to properly cite Eura et al. as the source of identifying the p53 125 peptide as a CD8 T cell HLA-A24 defined epitope; they list the reference only in a generic sense in the Introduction. Along those lines, they pick a panel of A24-binding epitopes (Fig 1) to test without any references (Figure 2).

Reviewer #3:

Remarks to the Author:

The manuscript, "Targeting mutant p53-Expressing Tumours with a T Cell Receptor-likeAntibody Specific for a Wild-type Antigen" by Low et al describes the production of an antibody that selectively targets mutant p53 peptide complexed to self-MHC. The antibody binds specifically to the target, mediates ADCC given its isotype, and selectively targets p53 mutant cells when conjugated to a cytotoxic drug. The work is convincing and well presented. However, its novelty is somewhat overstated by the authors. For example, host antibodies directed against mutant peptides such as EGFRvIII were described more than 25 years ago, and many groups have created engineered

antibodies targeting that mutated peptide. Creating an antibody that binds to the peptide in the context of self-MHC is an incremental advance, but it is a leap to conclude that this binding property makes an antibody into a T cell like molecule. The antibody has conventional structure, and does not specifically activate T cells. Its ADCC promotion properties are modest and are likely to have variable impact as the density of p53 mutant expression on various cancer cells will differ considerably. The capacity of the molecule to function as an ADC is interesting, but again the only potential advantage would be the tumor-specific target. Additional toxicity studies would be needed to convincingly demonstrate the selectivity and safety of ADCs targeting mutated p53 on tumors in the in vivo setting.

Reviewer#1:

Low et al. report the identification, isolation, engineering, biochemical and functional characterization of a T cell receptor-like (TCRL) monoclonal antibody (mAb) that recognizes the molecular complex of the p53-derived peptide p53125-134 bound to the human MHC-I molecule HLA-A24. Overall, the reagent described is novel, and the experiments well-performed. The characterization of the mAb suggests that this reagent deserves further study as a potential diagnostic or therapeutic agent in humans. Conceptually, the work is not novel, but follows an established approach of screening phage display libraries for Fab that bind a recombinant protein antigen, refining the phage library for identification of higher affinity reagents, and then engineering the Fab into an IgG1 to permit better characterization.

Although the paper is generally written clearly, several points need to be explained a little better in revision, and a number of detailed points that I will enumerate below. One nagging confusion is that the mAb is against the wildtype p53 sequence, and that the p53 mutants that are tested are mutants with respect to the function/expression of p53—thus, as I understand this, the mAb is detecting increased levels of the p53peptide/MHC complex resulting from dysregulation of the expression of the epitope rather than of the generation of a mutant epitope. This needs to be made clear from the start.

The review's understanding of the epitope that our engineered antibody targets is correct and we appreciate the reviewer's request for clarification in the manuscript. Hence, modifications to the abstract and introductions on pages 1, 3 and 4 have been made to emphasis and explain our approach of generating and engineering an antibody that recognizes a wild type p53 peptide in complex with the HLA*A24:02 MHC class I molecule but not a mutant epitope. This peptide sequence (amino acid 125-134) is rarely mutated among the vast majority of the mutant p53 proteins that have been reported (Clin Cancer Res 2000 6(6):2138-2145).

Specific points to be addressed in revision:

p. 1. Make clear that the mAb sees the WT p53 peptide in association with A24.

Modifications have been made to clarify the epitope that the antibody recognizes. Please see above.

p. 3. In the discussion of the mutants somewhere make it clear that the mAbs are not to tumor specific antigens.

Whilst wildtype p53 is not a tumour specific antigen, the aberrant expression of p53 protein levels due to mutations or transcriptional dysregulation led it to be recognized as an overexpressed tumour antigen. Our approach of targeting elevated mutant p53 proteins via our antibodies' recognition of a wildtype peptide sequence derived from these mutated proteins should hence qualify as an indirect strategy of targeting a tumour selective antigen. This point has been made clear in the manuscript.

p. 3. Period after oncogenesis.

Error has been addressed in manuscript.

p. 3. To remove mutant p53 in tumor cells (unclear).

Error has been addressed in manuscript.

p. 4. Degraded and presented as peptide-MHC antigens

Error has been addressed in manuscript.

p. 4. Define TCR-like antibody (TCRL) early on, please, not in a section heading.

We have defined TCR-like antibody (TCRL) in the introduction on page 4.

p. 6. The description that affinity was low, is never backed up quantitatively.

As mentioned in page 6, we hypothesized that the wildtype P1C1 antibody's affinity was low due to its inability to bind endogenously processed levels of p53₁₂₅₋₁₃₄/A24 complexes on the surface of the HT29 cells. This was subsequently backed up by the measured affinity of the germ line sequence reverted P1C1 (that displays similar binding avidity as the original P1C1, as shown in Fig. S1b), which was 116nM (Fig. S2e).

p. 6. pMHC antigen on “the” cell surface. Crucial, not pivotal.

Error has been addressed in manuscript.

p. 7. Individual clones “were” screened

Error has been addressed in manuscript.

p. 7. Due to “the” small theoretical diversity.

Error has been addressed in manuscript.

p. 7. And throughout—what precisely is the HLA-A*24 that is being used here? A*24:02

The HLA-A24 that we are targeting is HLA*A24:02 as defined on page 4. Similarly, for the transgenic mice, they harbour the $\alpha 1$ and $\alpha 2$ domains of the HLA*A24:02 gene as described on page 30.

p. 9. Affinity matured P1C1TM.....

throughout---although the mAb sees A*24 with peptide, there is no definitive experiment that shows that it fails to see the same peptide bound to a different allelic product MHC-I.

NetMHC prediction of p53₁₂₅₋₁₃₄ affinity for some of the more common alleles are as listed below:

Allele	Affinity (nM)
HLA*A24:02	26.16
HLA*A01:01	27481.29
HLA*A02:01	31978.31
HLA*A03:01	30502.81
HLA* B08:01	31156.92
HLA*B15:01	8571.88
HLA*B58:01	9347.40

As the predicted affinities are poor, synthesis of these recombinant pMHC complexes would not be feasible for the testing of the binding of the antibodies. Also, due to the poor affinity, the stable

complex of p53₁₂₅₋₁₃₄ peptide with any of these MHC class I molecules will probably be rare physiologically as well.

p. 13 and throughout...listing the pheno/genotype of the cell lines as p53R273H for instance, fails to emphasize that these are mutations that result in different levels of expression, and possibly different levels of presentation of the wild-type peptide.

We acknowledge that different mutations result in different levels of expression. We did find higher levels of antigen expression in cell lines that harbour mutant p53 proteins. This finding is presented in Figure S4e. More importantly, the effect of these mutations on the processing and presentation of p53 derived peptides is something that can be better examined with the use of antibodies such as ours that recognizes specific p53 derived pMHC complexes. We have looked at the effects of reagents that affect antigen processing and p53 degradation on p53 pMHC levels in Fig S6, but a follow up mechanistic study with cell lines that represent mutations in various domains of the p53 protein would provide a better understanding on the effects of these mutations on p53 stability and degradation.

p. 19 (BiTEs) target a cancer

Error has been addressed in manuscript.

p. 19 cells through the anti-CD3 arm

Error has been addressed in manuscript.

try to use TCRL throughout otherwise use TCR-like.....one or the other.

We have modified the manuscript to maintain consistency in the use of TCRL to refer to TCR-like antibodies in the manuscript.

p. 21 what is the affinity that is called "moderate"

Moderate affinity is an arbitrary term but we generally define it as K_D between 10 to 100 nM.

p. 23 The p53 move up to the next line.

Error has been addressed in manuscript.

p. 25 reference for Durocher et al?

We have included the missing reference for Durocher *et al* on page 25.

p. 28 equilibrium dissociation constants (KD), kinetic constants ka and kd in figure . also, what are the statistics of the curve fits for the binding parameters?

We have re-measured the binding kinetics and affinity of the antibodies by Biacore. The results are summarized in supplementary Fig. 2. The curve fit statistics are also included in the respective figures that are representative of the experimental replicates. All measurements had been repeated 2-4 times and the mean and SEM have been calculated and tabulated in supplementary Fig. 2e.

p. 29 what genotype of HLA-A*24 for the transgenics

The HLA*A24 allele in the transgenic mice is HLA*24:02.

p. 30 100 mm³

Error has been addressed in manuscript.

What is the journal's policy on MTA dependent provision of materials. Once reported the mab should be available for experimental purposes.

The Journal requires that all materials used to conduct experiments in the accepted manuscripts be made accessible to researchers through appropriate MTA between the respective institutes. We will follow this requirement and make our antibodies available to interested researchers once the manuscript is accepted and published.

p. 40. P1C1TM and an isotype control.....were conjugated.

Error has been addressed in manuscript.

p. 44. Figure 2. C. equilibrium constant K_D , kinetic constants k_a , and k_d . What are errors for the values?

See response above to the question (p.28) regarding the equilibrium and kinetic constant values and statistics.

p. 45. Figure 3. Mean fluorescence or MFI? How many times were these experiments done/

For Fig. 3a, the axes have been changed to specific MFI. MFI of the secondary antibody only control was subtracted from the individual samples to obtain the specific MFI that is presented in the figure. The data is representative of 2 independent experiments as stated on page 40.

p. 48 is growth inhibition the same as cytotoxicity?

We have edited the axes for Fig. 6 and Fig. S8 to % cytotoxicity for consistency. In the assays, we were comparing the number of viable cells post antibody drug conjugate treatment with respect to the untreated cells. Hence the data should be expressed as % cytotoxicity.

Suppl fig 2 same concerns for values and statistics as above.

See response above to the question regarding the equilibrium and kinetic constant values and statistics.

Comment on the peptide scan analysis. Since these seem to have been done only at one concentration, one should be careful not to over-interpret the results.

We acknowledge that the peptide scan analysis was done with only 1 concentration. However, we regard 10 μ M as a supraphysiological concentration of peptides and made the assumption that if P1C1TM does not stain cells pulsed with 10 μ M of peptides, it would not have significant affinity for them under physiological conditions.

Reviewer#2:

The article by Wang and collaborators describes the selection of an antibody specific for a p53/125-135 peptide complexed to HLA-A24, which mimics the reactivity of a T cell receptor. The strategy to isolate this antibody from a phage display library, followed by mutational affinity maturation, is well thought-out and well performed. Most of the experiments presented have appropriate controls and show that the affinity matured antibody is able to recognize a non-mutated p53 peptide only in tumor cells that express mutations in other regions of the p53 molecule, presumably because these tumors overexpress p53, or perhaps because the mutated p53 proteins are processed differently. Unfortunately, the authors do not distinguish the 2 possibilities, which could be done using this antibody, for example, by overexpression of WT p53 in an HLA-A24+ cell line.

We appreciate the Reviewer's suggestion and agree that the mentioned 2 possibilities can be differentiated by our antibody. To this end, instead of using artificial overexpression of WT p53 in a HLA-A24 cell line, we examined the level of p53 derived pMHCs on both resting and activated HLA-A24 T cells, for which the latter (activated T cells) showed elevated level of p53 protein in the cells. While our antibody virtually failed to bind to the resting T cells, it showed significant binding to the activated cells as evidenced by flow cytometry. This result suggested that the reason our antibody can recognize tumor cells could be due to higher level of p53 protein in the cells. We also performed experiments using proteasome inhibitor MG132 as well as treating the cells with interferon- γ and the p53/MDM2 interaction blocker nutlin, and came up with similar conclusion. These studies are presented in the Results and Discussion sessions in the manuscript.

In addition to the above issue, there are major points that need to be addressed:

1) The authors have failed to take into account that activated immune cells, in particular, T cells express elevated wt p53 and are potential targets for any probe targeting this antigen, especially one that has been engineered to be "high affinity" against a non-mutated p53 epitope. Supplemental Table 1 indicates "no significant findings" relating to toxicity of the probe. The toxicity studies in HLA-A24 transgenic mice have many caveats since the mouse p53 sequence in this epitope is different from and non-crossreactive with the human sequence (Fig. 3). At minimum, in vitro based experiments involving mitogen activated HLA-A24+ human T cells as targets are needed.

We appreciate the Reviewer's comment on activated T cells being potential targets of our antibody. We have conducted experiments to answer this question and the results are presented in Figures 4 and S5. To this end, we identified and isolated both HLA-A24+ and HLA-A24- PBMCs and saw no significant binding of P1C1TM on both sets of un-activated PBMCs. We further purified T cells and assessed binding of P1C1TM on activated and resting purified T cells. In our hands, activation of T cells with anti-CD3 and anti-CD28 polymers upregulated p53 protein expression and indeed resulted in elevated levels of p53 derived pMHC recognized by our P1C1TM antibody staining. It has been reported (Immunity 2014 40(5):681-691) that p53 upregulation in CD4 T cells is induced by IL-2 signalling whilst TCR signalling resulted in downregulation of p53. Antigen and IL-2 stimulation thus resulted in a net transient increase in p53 within 48hrs that rapidly fell to baseline by 96hrs. Thus, we hypothesize that resting T cells and PBMCs more accurately represent healthy tissues wherein physiological levels of p53 expression does not result in significant levels of p53 derived pMHCs that would result in their being targeted by P1C1TM. Only when p53 expression are elevated, e.g. in the case of activated T cells, p53 derived pMHCs are elevated albeit transiently and could potentially become targets of P1C1TM. In the therapeutic setting, by ADC for example, we believe that the targeting and subsequent killing of only the activated T cells likely would not pose a toxicity concern. On the other hand, tumour cells harbouring p53 mutations constitutively present high levels of p53 derived pMHCs and thus are more likely targets for P1C1TM binding/killing. We also believe that,

with better selection of toxic payload and ADC linker/spacer design as well as further affinity tuning of P1C1TM, it is possible to better discriminate the activated T cells from tumour cells.

Regarding the use of the HLA-A24 transgenic mouse for our toxicity studies, the reviewer has rightfully pointed out that the model bears several caveats as the mouse p53 sequence differs from our targeted p53₁₂₅₋₁₃₄ sequence by 2 amino acids and is not recognized by our antibody P1C1. Nonetheless, this study was intended to evaluate off-target toxicities on a whole-proteome scale that could arise from cross reactivity of our antibody to other HLA-A24 pMHC epitopes presented *in vivo* in the mouse. The fact that only very minimal aberrant pathology was observed in this mouse model suggested minimal off-target toxicity concerns by our antibody.

2) The authors assume that because tumors that do not have p53 mutations, they do not react with the antibody, normal cells will also not react with the antibody. However, many tumors express lower levels of MHC-I as compared to normal tissues, so peptide/MHC density could play a role in potential reactivity in normal tissues. Can the antibody be used in histological sections to stain tumors (with and without p53 mutations) and normal tissue samples in HLA-A24 individuals? These experiments could provide additional information regarding potential off-target toxicities.

We have attempted immunohistochemistry (IHC) experiments with the positive control HT29 cells. Unfortunately, we were unable to detect p53₁₂₅₋₁₃₄/A24 pMHCs on the surface of HT29 cells. This is likely because the MHC/peptide complex falls apart under the harsh IHC sample processing conditions. The antibody is thus unsuitable for this application. Hence, we are unable to assess the off-tissue cross-reactivity on tissue microarrays (TMA) with P1C1 by IHC. However, Zhu *et al* (J Immunol 2006; 176:3223-3232) engineered a soluble TCR multimer specific for the p53₂₆₄₋₂₇₂/HLA* A0201 pMHC and assessed levels of p53 pMHCs presented on normal tissue arrays in a human TMA. Encouragingly, low or no staining of the TCR multimer was observed in almost all the normal tissues. The authors then stained human tumor sections with the TCR multimers and saw that 49% of colorectal cancer and 34% of breast cancer sections were positively stained, of which, 88% and 94% of the sections respectively are positive for HLA-A2. Hence this study provides further support of the specificity of p53 derived pMHC complexes as a tumour specific antigen.

3) In the experiment with results shown in Fig. 7, why did the authors limit treatment to a single dose of antibody? Multiple doses (e.g., weekly) could have provided a better therapeutic effect. Also, experiments with minimal residual disease (treating at an earlier time point) or a “metastatic” model intravenous injection monitoring tumor growth in liver/lungs with bioluminescence would greatly improve the message of therapeutic potential of the antibody.

We carried out additional *in vivo* efficacy experiments as suggested by the reviewer. We explored a regime whereby mice were first treated with a loading dose of 1mg/kg followed by additional weekly doses of 0.3mg/kg for 2 subsequent weeks. We did not see additional benefits in terms of tumour growth control. However, we did notice that mice started to become unwell after the third dose and had to be sacrificed. We hypothesized that the accumulation of the potent PNU could be the cause of their health status. As in a response to previous question about, our experiments using PNU ADC in the mouse models only served as a proof concept. A better therapeutic window can be achieved through optimization in ADC design and engineering.

Also, we appreciate the Reviewer’s suggestion and explored a metastatic model in which the HT29 tumor cells were injected in NSG mice intravenously. We monitored the fitness and survival of mice treated with either our antibody drug conjugate (ADC) or the unconjugated antibody and saw that mice treated with a single dose of 1mg/kg of ADC had a statistically significant survival advantage

over the control treated mice (Fig. 7c). The results further support the therapeutic potential of the antibody.

4) Although the use of xenograph-based experiments does demonstrate the efficacy of the probe, such “cutting edge experiments” now require the use of orthotopic xenografts derived from specimens obtained from patients, which have been characterized relative to potential defects in antigen processing and presentation machinery, which generally are commonly found in human tumors.

We are currently in the midst of establishing the patient derived xenograft (PDX) models to assess our P1C1TM antibody and look forward to reporting our findings in a future publication.

5) The manuscript has too much data presented, in both the text figures and supplemental figures; there are too many tumor cell lines and variables involving A24 and p53 status to keep straight. The whole paper should be streamlined. While the panel of tumor cell lines utilized is extensive, the use of any target cell, whether naturally presenting A24 or, more importantly, transfected to express A24 points out that the levels of A24 expression and defects in antigen processing and presentation (no evidence of levels of efficacy of transfection in these cell lines is indicated), need to be understood to relate the specificity of the probe. Moreover, there is 1) no mention of any potential shared HLA specificities among the non-transfected positive and negative tumor cell lines used that also might be capable of presenting the epitope, and 2) no mention of the levels of epitope that are naturally presented by tumors that are recognized by T cells to get a better understanding of the affinity of the probes.

We feel that it is necessary to use an appropriate panel of cell lines to prove that P1C1TM binding was specific to the intended epitope across cell lines of various origins and different p53 mutational status, hence a large volume of data is presented to the readers who may otherwise have doubts. Furthermore, we agree with the Reviewer that the levels of the A24 expression in the transfected cell lines need to be understood. To this end, we purified the A24-transduced cells by sorting for HLA-A24 expression and assays were conducted only when >95% cells had HLA-A24 expression. In Fig. 4b and Fig. 4c, inserts show P1C1TM staining of cells pulsed with 10 μ M of exogenous peptides. The data well illustrates the purity of transduced cells as well as the relative levels of expression of HLA-A24.

Regarding the question of shared HLA specificities, we believe that the cells used in our experiments do not have known shared HLA alleles as tabulated by the TRON cell line portal (<http://celllines.tron-mainz.de/>). And also as mentioned above in response to reviewer 1's question, the predicted affinities of the peptide to various common alleles are extremely low. Hence, we hypothesize that there are very low chances of physiologically stable complexes of the p53₁₂₅₋₁₃₄ peptide with other alleles.

Lastly, to assess the levels of epitope expression, we have labelled our antibody with PE and attempted to quantify the levels of epitope on the surface of various cell lines using the BD Quantibrite™ PE Fluorescence Quantification Kit. The degree of labelling was around 1-1.3 PE per antibody. The levels of fluorescence we obtained in these experiments were approaching the lower limit of the quantification standard curve, hence the calculated copy number of the epitope may not be accurate. Nevertheless, the data suggested low level of antigens expression in general, as expected, and we estimate that the levels fall in the 10² - 10³ range for the wildtype and mutant p53-expressing cells. We are currently working with our colleague to produce radioisotope labelled antibody for a more sensitive and accurate measurement and will report the results in the future publication.

Minor Comments:

1) Many of the figures are not as well defined as they could be. The use of APC to denote the target cells (Figure 1) rather than the actual tumor cell is confusing. In many figures the descriptions of the cell lines vary from one figure to the next and/or the text. Sometimes the p53 mutational site is given, other times it "p53 mut", "p53 pos" or p53.

We have reformatted the figures to standardize the labelling of the p53 mutation status and better clarify the axes.

2) The authors fail to properly cite Eura et al. as the source of identifying the p53 125 peptide as a CD8 T cell HLA-A24 defined epitope; they list the reference only in a generic sense in the Introduction. Along those lines, they pick a panel of A24-binding epitopes (Fig 1) to test without any references (Figure 2).

We have included the appropriate references for the peptides that were used in the studies.

Reviewer#3:

The manuscript, "Targeting mutant p53-Expressing Tumours with a T Cell Receptor-like Antibody Specific for a Wild-type Antigen" by Low et al describes the production of an antibody that selectively targets mutant p53 peptide complexed to self-MHC. The antibody binds specifically to the target, mediates ADCC given its isotype, and selectively targets p53 mutant cells when conjugated to a cytotoxic drug. The work is convincing and well presented. However, its novelty is somewhat overstated by the authors. For example, host antibodies directed against mutant peptides such as EGFRvIII were described more than 25 years ago, and many groups have created engineered antibodies targeting that mutated peptide. Creating an antibody that binds to the peptide in the context of self-MHC is an incremental advance, but it is a leap to conclude that this binding property makes an antibody into a T cell like molecule. The antibody has conventional structure, and does not specifically activate T cells. Its ADCC promotion properties are modest and are likely to have variable impact as the density of p53 mutant expression on various cancer cells will differ considerably. The capacity of the molecule to function as an ADC is interesting, but again the only potential advantage would be the tumor-specific target. Additional toxicity studies would be needed to convincingly demonstrate the selectivity and safety of ADCs targeting mutated p53 on tumors in the in vivo setting.

First, we want to clarify that this study describes the engineering of a "T cell receptor-like", but not a "T cell-like", antibody. "T cell receptor-like" antibodies, such as our antibody P1C1, mimic the specificity of the T cell receptor for the recognition of a peptide-MHC complex. Such antibodies in their conventional structure are unable to activate T cells, and we similarly do not claim or imply that our antibody can. However, antibodies can be engineered into formats that can have the potential to activate T cells upon antigen engagement. The bispecific T cell engager (BiTE) antibodies and chimeric antigen receptor (CAR) T cells are areas that we are currently exploring in our lab. Also, we want to further clarify that, unlike the EGFRvIII example brought up by the reviewer, the antibody described in this work targets the MHC complex of a wildtype p53 peptide sequence that is present in both the wildtype and the vast majority of the p53 mutant proteins. The principle of potential applications of our antibody is based on the differential expression of the epitope between normal cells (low or even undetectable) and tumor cells (higher expression).

In terms of the ADCC results, we agree with the reviewer that the potency is modest and we think this is likely due to the low density of p53₁₂₅₋₁₃₄/HLA-A24 pMHCs as pointed out by the reviewer. In our study, we are exploring the prospect of target tumour cells based on the expression of mutant p53 by the fact that there are potentially different levels of pMHC on the surface of healthy cells expressing wildtype p53 and malignant cells expressing elevated levels of mutant p53. Hence, we looked at the option of using the antibody as an ADC, leveraging on not only the differential levels of pMHC on tumour cells but also the rapid internalization of the pMHC to facilitate the cytotoxic payload mediated cell death. Mutations in p53 is very common in most tumours, hence demonstrating the potential to target these tumour cells via the pMHC could be a significant step in developing tumour-selective therapeutic strategies.

As for the toxicity, we agree with the reviewer that it is the utmost important issue and we have addressed the questions that were also raised by Reviewer 2. We performed toxicity studies using healthy donor PBMCs. We saw that activated T cells upregulate p53 and present higher levels of p53 derived pMHCs. We did see cytotoxicity of these activated T cells with high doses of our ADCs but this could be due to T cells' increased sensitivity to the PNU drug. Better design of the ADC could potentially help mitigate these unwanted toxicities. In addition, we assessed the non-specific off-target toxicities of the ADC by using transgenic HLA*A24:02 mice, of which we saw very minimal of. However, as our antibodies do not recognize the corresponding murine p53-derived peptide sequence in complex with HLA*A24:02, proper on-target in vivo toxicity studies will have to rely on a double transgenic human p53/HLA*A24:02 expressing mouse model, which is currently unavailable. Apart from the studies in cell lines, we are currently establishing xenograft models with samples derived from cancer patients. We look forward to sharing the results in the follow up studies.

We hope that these additional studies sufficiently address the concerns highlighted and provide further support to its consideration for publication by Nature Communications.

Sincerely,

Cheng-I Wang

REVIEWERS' COMMENTS:

Reviewer #1 (Remarks to the Author):

Low et al have revised the manuscript to address most of the concerns of this reviewer and those of the other reviewers as well. There remains some confusion about the specificity of the antibody in the way the paper is presented. The antibody sees the wild-type P53125-134 sequence, characterized carefully with peptide substitutions, etc. What remains confusing is that the cell lines/tumors that have high reactivity with the affinity matured mAb (P1C1TM) are repeatedly called those with "mutant P53". This is accurate, but because the text doesn't emphasize that the mutant P53s are in residues of the protein that are not included in the antigenic peptide, the reader may be confused about an antibody against the wild type peptide/MHC complex reacting with "mutant P53" expressing cells. It would be helpful if at least in early mention of the mutant P53-expressing cells the author would emphasize that the mutation does not involve the antigenic peptide epitope.

Additionally:

Line 90, indicating that cells "were stained" and then in the following sentence, "minimal staining" was seen. This is confusion. Perhaps, "cells were examined for their ability to be stained...." Or "cells were tested for their ability to bind the P1C1TM antibody. Would be clearer.

Line 168. Cell lines not cells lines.

Line 247. Make it clear that L234/L235 refer to the antibody residues.

Reviewer #2 (Remarks to the Author):

The authors have generally addressed the concerns of this reviewer in their resubmission. There is still a minor issue relative to figure 4d in that the term "APC"

rather than "PBMC" was used and the abbreviation of the secondary Ab,

anti-human IgG-AF647, should be added to the text on line 558.

Reviewer #3 (Remarks to the Author):

The authors have satisfactorily addressed my concerns.

Dr Cheng-I Wang
Singapore Immunology Network (SigN)
8A Biomedical Grove
Immunos Level 3
Singapore 138648
DID: +65 6407 0083

Dear Reviewers,

We like to thank you for your time and effort in reviewing our manuscript.

We have appreciated the comments and suggestions and would like to address the points as highlighted by the individual reviewers below;

Reviewer #1 (Remarks to the Author):

Low et al have revised the manuscript to address most of the concerns of this reviewer and those of the other reviewers as well. There remains some confusion about the specificity of the antibody in the way the paper is presented. The antibody sees the wild-type P53₁₂₅₋₁₃₄ sequence, characterized carefully with peptide substitutions, etc. What remains confusing is that the cell lines/tumors that have high reactivity with the affinity matured mAb (P1C1TM) are repeatedly called those with "mutant P53". This is accurate, but because the text doesn't emphasize that the mutant P53s are in residues of the protein that are not included in the antigenic peptide, the reader may be confused about an antibody against the wild type peptide/MHC complex reacting with "mutant P53" expressing cells. It would be helpful if at least in early mention of the mutant P53-expressing cells the author would emphasize that the mutation does not involve the antigenic peptide epitope.

We have clarified in lines 52-57 that our hypothesis is that by taking advantage of the fact that tumours expressing mutant p53 have elevated levels of intracellular p53 proteins and thus with continual degradation, will result in increased levels of p53 derived peptides presented on the tumour cell surface as pMHCs, such pMHCs presenting peptides of unmutated or wild-type sequences (which should statistically exist in greater abundance as compared to peptides harbouring a mutation), are potential markers to differentiate healthy cells expressing low levels of wild-type p53 proteins from malignant cells expressing high levels of mutant p53.

We further described the p53 mutation status of the cell line used in the manuscript in the cell lines and reagents subsection of the Methods section. None of them harbour mutations that lie in the sequence of our peptide of interest, i.e. p53₁₂₅₋₁₃₄.

Additionally:

Line 90, indicating that cells "were stained" and then in the following sentence, "minimal staining" was seen. This is confusion. Perhaps, "cells were examined for their ability to be stained..." Or "cells were tested for their ability to bind the P1C1TM antibody. Would be clearer.

We have edited accordingly the highlighted sentence to "Lastly, the ability of P1C1 to recognize and bind HLA-A24⁺ HT29 cells that express high levels of mutant p53 was examined. However, minimal staining was seen even with 10µg mL⁻¹ of antibodies." (Lines 92-95) to avoid confusion.

Line 168. Cell lines not cells lines.

We have edited the sentence "Three HLA-A24+ cells lines" to "Three HLA-A24+ cell lines" (Line 172).

Line 247. Make it clear that L234/L235 refer to the antibody residues.

We have further elaborated that "The substitutions of leucine 234 and leucine 235 to alanines (LALA) in the antibody Fc region" (Line 248-249).

Reviewer #2 (Remarks to the Author):

The authors have generally addressed the concerns of this reviewer in their resubmission. There is still a minor issue relative to figure 4d in that the term "APC" rather than "PBMC" was used and the abbreviation of the secondary Ab, anti-human IgG-AF647, should be added to the text on line 558.

The term APC (allophycocyanin) was used to refer to the channel in which we detected the A24 expression using the AF647 conjugated anti-A24 antibodies and not the antigen presenting cells in PBMCs. We have removed it from Figure 4d to prevent confusion.

The missing abbreviation "**anti-human IgG-AF647**" has been added as pointed out.

We hope that these additional studies are sufficiently address the concerns highlighted and provide further support to its consideration for publication by Nature Communications.

Sincerely,

Cheng-I Wang